# Efficient Action-Constrained Reinforcement Learning via Acceptance-Rejection Method and Augmented MDPs

**Wei Hung[1]   Shao-Hua Sun[2]   Ping-Chun Hsieh[1]**
[1]National Yang Ming Chiao Tung University, Hsinchu, Taiwan
[2]National Taiwan University, Taipei, Taiwan
{hwei1048576.cs08, pinghsieh}@nycu.edu.tw

## Abstract

Action-constrained reinforcement learning (ACRL) is a generic framework for learning control policies with zero action constraint violation, which is required by various safety-critical and resource-constrained applications. The existing ACRL methods can typically achieve favorable constraint satisfaction but at the cost of either a high computational burden incurred by the quadratic programs (QP) or increased architectural complexity due to the use of sophisticated generative models. In this paper, we propose a generic and computationally efficient framework that can adapt a standard unconstrained RL method to ACRL through two modifications: (i) To enforce the action constraints, we leverage the classic acceptance-rejection method, where we treat the unconstrained policy as the proposal distribution and derive a modified policy with feasible actions. (ii) To improve the acceptance rate of the proposal distribution, we construct an augmented two-objective Markov decision process (MDP), which includes additional self-loop state transitions and a penalty signal for the rejected actions. This augmented MDP incentivizes the learned policy to stay close to the feasible action sets. Through extensive experiments in both robot control and resource allocation domains, we demonstrate that the proposed framework enjoys faster training progress, better constraint satisfaction, and a lower action inference time simultaneously than the state-of-the-art ACRL methods. We have made the source code publicly available[*] to encourage further research in this direction.

## 1 Introduction

Action-constrained reinforcement learning (ACRL) aims to find an optimal policy maximizing expected cumulative return while satisfying constraints imposed on the action space and has served as a generic framework for learning sequential decision making in both safety-critical and resource-constrained applications. As a classic example, robot control is usually subject to the inherent kinematic constraints of the robots, *e.g.*, torque or output power, which need to be satisfied throughout the training and the inference stages to avoid damage to physical components (Singletary et al., 2021; Tang et al., 2024; Liu et al., 2024). Another example is dynamic resource allocation for networked systems (Chen et al., 2023; Jay et al., 2019; Chen et al., 2021), such as communication networks and bike sharing systems (Zhang et al., 2022; 2021), which involve the capacity constraints on the communication links and the docking facilities, respectively. To prevent network congestion and resource over-utilization, these constraints must be considered at each step of training and deployment. Given its wide applicability, developing practical ACRL algorithms that can learn policies accruing high returns with minimal constraint violation is essential.

Existing ACRL methods have explored the following techniques: (i) *Action projection*: As a conceptually simple and widely-used technique, action projection finds a feasible action closest to the original unconstrained action produced by the policy. The projection step can be used in action post-processing (Kasaura et al., 2023) or implemented by a differentiable projection layer (Amos

---

[*]https://github.com/NYCU-RL-Bandits-Lab/ARAM

& Kolter, 2017) as part of the policy network of a standard deep RL algorithm for end-to-end training (Pham et al., 2018; Dalal et al., 2018; Bhatia et al., 2019). Despite the simplicity, to find close feasible actions, action projection needs to solve a quadratic program (QP), which is computationally costly and scales poorly to high-dimensional action spaces (Ichnowski et al., 2021). (ii) *Frank-Wolfe search*: Lin et al. (2021) propose to decouple policy updates from action constraints by a Frank-Wolfe search subroutine. Despite its effectiveness, Frank-Wolfe method requires solving multiple QPs per training iteration and therefore suffers from substantially higher training time. (iii) *Generative models*: To replace the projection step, generative models, such as Normalizing Flows (Kobyzev et al., 2020), have been employed as a learnable projection layer that is trained to satisfy the constraints (Brahmanage et al., 2023; Chen et al., 2023).

Despite the recent advancement in ACRL, existing algorithms suffer from significant computational overhead induced by solving QPs or learning sophisticated models, such as Normalizing Flows. Table 1 summarizes the limitations of existing methods in interaction time, training load, and action violation rate. This motivates us to develop a more efficient ACRL approach.

Table 1: A qualitative comparison of ACRL algorithms.

| Algorithm | Action violation rate | Generative model | Interaction time | Training load | Remarks |
|---|---|---|---|---|---|
| OptLayer (Pham et al., 2018) | High | ✗ | High | **Low** | Zero-gradient issue |
| ApprOpt (Bhatia et al., 2019) | High | ✗ | High | **Low** | Zero-gradient issue |
| NFWPO (Lin et al., 2021) | **Low** | ✗ | **Low** | High | Rely heavily on QPs |
| FlowPG (Brahmanage et al., 2023) | **Low** | ✓ | **Low** | High | Require pre-training |
| DPre+ (Kasaura et al., 2023) | High | ✗ | High | **Low** | - |
| SPre+ (Kasaura et al., 2023) | High | ✗ | High | **Low** | - |
| IAR-A2C (Chen et al., 2023) | **Low** | ✓ | **Low** | High | Support only discrete actions |
| ARAM (Ours) | **Low** | ✗ | **Low** | **Low** | - |

We propose a framework called ARAM, which augments a standard deep RL algorithm with two modifications: **A**cceptance-**R**ejection method and **A**ugmented **M**DPs. (i) *Acceptance-rejection method*: Given a policy network, ARAM enforces the action constraints by rethinking ACRL through acceptance-rejection sampling, *i.e.*, first sampling actions from the unconstrained policy and then only accepting those that are in the feasible action set. This sampling strategy can substantially reduce the need for solving QPs, compared to the methods built on the action projection step or the Frank-Wolfe search. (ii) *Augmented unconstrained two-objective Markov decision process*: One technical issue of the acceptance-rejection method is the possibly low acceptance rate, which is likely to occur in the early training stage. Under a low acceptance rate, the sampling process could take excessively long and thereby incur a high training overhead. To improve the acceptance rate, we augment the original MDP with additional self-loop transitions and a penalty function induced by the event whether the action is accepted. Through this augmented MDP, we leverage the penalty induced by constraint violation to guide the policy distribution towards regions of higher acceptance rate. Notably, these two modifications can be combined with any standard deep RL algorithm. In this paper, we take the Soft Actor Critic (SAC) as the base RL algorithm. Moreover, to obviate the need for hyperparameter tuning of the penalty weight, we directly leverage the multi-objective extension of SAC that can learn policies under all the penalty weights.

We evaluate the proposed ARAM in various ACRL benchmarks, including the MuJoCo locomotion tasks and resource allocation of communication networks and bike sharing systems. The experimental results show that: (i) ARAM enjoys faster learning progress than the state-of-the-art ACRL methods, measured either in environment steps or wall clock time. Moreover, the difference is particularly significant under wall clock time thanks to the low training time of our design. (ii) ARAM requires significantly fewer QP operations than the other ACRL benchmark methods, mostly by 2-5 orders of magnitude fewer. (iii) ARAM indeed achieves high action acceptance rate through the guidance of the augmented MDP. (iv) ARAM also enjoys the lowest per-action inference time as it largely obviates the need for QP operations and learns without using a generative model.

## 2 RELATED WORK

**Action-Constrained RL.** The first category focuses on ensuring that the actions meet the constraints at each step of the training and evaluation processes. To ensure zero constraint violation, one nat-

ural and commonly-used technique is *action projection*, which finds a feasible action closest to the original unconstrained action produced by the policy. To enable end-to-end training, multiple ACRL methods have incorporated the *differentiable projection layer* (Amos & Kolter, 2017) as the output layer of the policy network, such as OptLayer (Pham et al., 2018), Safety Layer (Dalal et al., 2018), and Approximate OptLayer (Bhatia et al., 2019). However, these projection layers are known to suffer from slow learning in ACRL due to the zero-gradient issue (Lin et al., 2021).

To address the zero-gradient issue, several approaches have been adopted recently: (i) *Frank-Wolfe search*: Lin et al. (2021) propose Neural Frank-Wolfe Policy Optimization (NFWPO), which decouples policy updates from action constraints by a Frank-Wolfe search subroutine. Despite its effectiveness, Frank-Wolfe search involves solving multiple quadratic programs (QP) per training iteration, and this is computationally very costly and is known to scale poorly to high-dimensional action spaces (Ichnowski et al., 2021). (ii) *Removing projection layers and using pre-projected actions for critic learning*: Without using differentiable projection layers, Kasaura et al. (2023) propose DDPG-based DPre+ and SAC-based SPre+, which use a QP-based projection step for action post-processing, train the critic with pre-projected actions, and apply a penalty term to reduce constraint violation. Notably, DPre+ and SPre+ achieve fairly strong reward performance but still at the cost of high computational overhead incurred by QPs. (iii) *Generative models*: To replace the projection layer, generative models, e.g., normalizing flow, have been integrated into the policy network, such as FlowPG for continuous control (Brahmanage et al., 2023) and IAR-A2C for discrete control (Chen et al., 2023), to generate multi-modal action distributions that can better satisfy the constraints. Despite the effectiveness, learning a sophisticated generative model adds substantial design complexity to ACRL. By contrast, the proposed ARAM largely removes the overhead of QPs and completely obviates the need for generative models.

**RL for Constrained MDPs.** The other class of methods focuses on ensuring the long-term average action safety by defining a cost function and modeling the problem as a Constrained MDP (CMDP) (Altman, 2021). For example, Constrained Policy Optimization (CPO) (Achiam et al., 2017) is the first policy gradient method developed to solve CMDPs. It uses the Fisher information matrix and second-order Taylor expansion to ensure safety constraints, but it is computationally expensive and requires more samples, potentially reducing efficiency. To address these, Tessler et al. (2019) propose Reward Constrained Policy Optimization (RCPO), which leverages primal-dual methods to improve both the efficiency and effectiveness of policy optimization under constraints. Building on similar goals of improving efficiency, FOCOPS (Zhang et al., 2020) takes a different approach by using a first-order approximation for policy optimization. This reduces computational complexity but introduces convergence issues due to approximation errors in the first-order constraints. The above list is by no means exhaustive and is only meant to provide an overview of this line of research. Please refer to (Liu et al., 2021) for more related prior works. While these methods can ensure that the long-term expected cost remains under a certain threshold, they fail to enforce action constraints at every environment step needed in ACRL throughout training and deployment.

**Augmented Safety Mechanisms in RL.** To improve safety in exploration, Eysenbach et al. (2018) introduced a reset framework, which detects when the agent enters unsafe states and resets it, thereby improving both safety and sampling efficiency. Building on this, Thananjeyan et al. (2021) introduced the concept of learned recovery zones, ensuring that when the agent deviates from safe limits, it can autonomously return to a safe state, providing more robust safety guarantees during exploration. In parallel, Thomas et al. (2021) proposed a specialized Markov Decision Process (MDP) to constrain the training process, helping to further avoid unsafe actions during exploration. In addition to mechanisms that correct unsafe actions, Safety Augmented Value Estimation from Demonstrations (SAVED) (Thananjeyan et al., 2020) employs model predictive control (MPC) to proactively avoid unsafe actions by updating the policy, specifically preventing infeasible actions that could lead to dangerous situations. Complementing this, Yu et al. (2022) proposed Safety Editor (SEditor), a mechanism that transforms actions produced by a utility maximizer into safe alternatives, preventing violations of safety constraints during execution. While these methods can not fully guarantee that the policy's actions always remain within safe regions, they offer an effective approach to significantly reduce unsafe behaviors, laying a foundation to develop more robust safety mechanisms.

## 3 PRELIMINARIES: ACTION-CONSTRAINED REINFORCEMENT LEARNING

In ACRL, we consider an action-constrained Markov Decision Process (MDP). Given a set $\mathcal{X}$, let $\Delta(\mathcal{X})$ denote the set of all probability distributions on $\mathcal{X}$. An action-constrained MDP is defined by a tuple $\mathcal{M} := (\mathcal{S}, \mathcal{A}, \mathcal{P}, \gamma, r, \mathcal{C})$, where $\mathcal{S}$ denotes the state space, $\mathcal{A}$ denotes the action space, $\mathcal{P} : \mathcal{S} \times \mathcal{A} \to \Delta(\mathcal{S})$ serves as the transition kernel, $\gamma \in (0, 1)$ is the discount factor, $r : \mathcal{S} \times \mathcal{A} \to \mathbb{R}$ denotes the bounded reward function. Without loss of generality, we presume the reward $r(s, a)$ to lie in the $[0, 1]$ interval since we can rescale a bounded reward function to the range of $[0, 1]$ given the maximum and minimum possible reward values. For each $s \in \mathcal{S}$, there is a non-empty feasible action set $\mathcal{C}(s) \subseteq \mathcal{A}$ induced by the underlying collection of action constraints. That is to say, no actions outside the feasible set $\mathcal{C}(s)$ can be applied to the environment, ensuring that only valid actions are considered within the system dynamics. Notably, we make no assumption on the structure of $\mathcal{C}(s)$ (and hence $\mathcal{C}(s)$ needs not be convex).

At each time $t \in \mathbb{N} \cup \{0\}$, the learner observes the current state $s_t \in \mathcal{S}$ of the environment, selects a feasible action $a_t \in \mathcal{C}(s_t)$, and receives reward $r_t$. We use $\pi : \mathcal{S} \to \Delta(\mathcal{A})$ to denote a Markov stationary stochastic policy, which is updated iteratively by the learner. Given a policy $\pi$, the Q functions $Q(\cdot, \cdot; \pi) : \mathcal{S} \times \mathcal{A} \to \mathbb{R}$ is defined as $Q(s, a; \pi) := \mathbb{E}\left[\sum_{t=0}^{\infty} \gamma^t r_t | s_0 = s, a_0 = a; \pi\right]$, which can be characterized as the unique solution to the following Bellman equation:

$$Q(s, a; \pi) = r(s, a) + \gamma \mathbb{E}_{s' \sim \mathcal{P}, a' \sim \pi(\cdot|s')}[Q(s', a'; \pi)]. \tag{1}$$

To learn a policy and the corresponding Q function under large state and action spaces, we use the parameterized functions $\pi_\phi : \mathcal{S} \to \Delta(\mathcal{A})$ and $Q_\theta : \mathcal{S} \times \mathcal{A} \to \mathbb{R}$ as function approximators, where $\phi$ and $\theta$ typically denote the parameters of neural networks in the deep RL literature. Our goal is to learn an optimal policy $\pi^*$ such that $Q(s, a; \pi^*) \geq Q(s, a; \pi)$, for all $s \in \mathcal{S}, a \in \mathcal{C}(s)$ and $\pi \in \Pi_\mathcal{C}$, where $\Pi_\mathcal{C} := \{\pi : \mathcal{S} \to \Delta(\mathcal{C})\}$, which denotes the set of all feasible policies. We also use $\Pi_\mathcal{A} := \{\pi : \mathcal{S} \to \Delta(\mathcal{A})\}$ to denote the set of all unconstrained Markov stationary policies.

**Notations.** Throughout the paper, we use $\langle \boldsymbol{x}, \boldsymbol{y} \rangle$ to denote the inner product of two real vectors $\boldsymbol{x}, \boldsymbol{y}$. Moreover, we use $\mathbf{1}_d$ to denote the $d$-dimensional vector of all ones.

## 4 ALGORITHM

To address ACRL, we devise and introduce two main modifications to existing deep RL algorithms: the acceptance-rejection method (Section 4.1) and the augmented two-objective MDP (Section 4.2). Then, we describe a practical multi-objective RL implementation of ARAM in Section 4.3.

### 4.1 ACCEPTANCE-REJECTION METHOD

To adapt a standard deep RL method to ACRL, we need to convert an unconstrained policy $\pi_\phi \in \Pi_\mathcal{A}$ into a feasible policy $\pi_\phi^\dagger \in \Pi_\mathcal{C}$. Notably, the action constraints, in general, can be complex and take arbitrary forms of expression. As a result, the feasible action sets $\mathcal{C}(s)$ are likely to be rather unstructured. To tackle this, we propose to rethink the constraint satisfaction in ACRL through the classic acceptance-rejection method (ARM), which is a generic algorithm for sampling from general distributions (Kroese et al., 2013).

**Using ARM in the context of ACRL.** For didactic purposes, here we focus on the continuous control and assume that the action space $\mathcal{A}$ is a compact convex set despite that the same argument can work seamlessly under discrete action spaces. Let $f$ and $g$ be two probability density functions over $\mathcal{A}$. ARM can generate random variables that follow the target distribution $f$ while drawing samples from another *proposal distribution* $g$. To put ARM in the context of ACRL, let us fix a state $s$ and take the constrained and the unconstrained policies as the target and the proposal distributions, respectively, *i.e.*, $f \equiv \pi_\phi^\dagger(s) \in \Delta(\mathcal{C}(s))$, $g \equiv \pi_\phi(s) \in \Delta(\mathcal{A})$. Clearly, we have $\pi_\phi^\dagger(a|s) = 0$ for all $a \notin \mathcal{C}(s)$. Let $M > 0$ be some constant such that $M \cdot \pi_\phi(a|s) \geq \pi_\phi^\dagger(a|s)$, for all $a \in \mathcal{A}$. ARM can generate an action that follows $\pi_\phi^\dagger(s)$:

1. Generate $a' \in \mathcal{A}$ from the unconstrained $\pi_\phi(s)$.

2. If $a' \in \mathcal{C}(s)$, accept $a'$ with probability $\pi_\phi^\dagger(a'|s)/(M \cdot \pi_\phi(a'|s))$; otherwise, if $a' \notin \mathcal{C}(s)$, reject $a'$ and return to the first step.

**Choices of target distribution $\pi_\phi^\dagger$.** Note that we have the freedom to configure the desired $\pi_\phi^\dagger$ for specific purposes, *e.g.*, exploration. One convenient choice is to simply set $\pi_\phi^\dagger(a|s) \propto \pi_\phi(a|s)$, for all $a \in \mathcal{C}(s)$. In this case, we can always accept an action $a' \in \mathcal{C}(s)$ in the above step 2 if we set $M = 1/(\int_{a \in \mathcal{C}(s)} \pi_\phi(a|s)da)$.

**Salient features of ARM.** The advantages of using ARM in ACRL are two-fold: (i) *Efficiency*: ARM is computationally efficient as it only requires checking if an action satisfies the constraints. As a result, ARM largely obviates the need to solve QPs or learn a generative model. (ii) *Generality*: ARM is very general, *i.e.*, can be integrated with any standard unconstrained RL method.

**Issues of low acceptance rate under ARM.** Despite the efficiency and generality of ARM, solely naively adopting ARM in ACRL can lead to keeping sampling invalid actions and getting stuck when the policy has a low action acceptance rate. The issues are two-fold: (i) The ARM procedure could repeat indefinitely with a near-zero acceptance probability. In ACRL, this scenario is likely to happen at the early training stage since the randomly initialized action distribution can be drastically different from the feasible action set. (ii) A low acceptance rate typically implies a poor action coverage of the policy over $\mathcal{C}(s)$. This would significantly affect the performance in the cumulative reward. To address this issue by increasing the action acceptance rate through the course of learning, we present our solution in Section 4.2.

## 4.2 Augmented Unconstrained Two-Objective MDP

To mitigate this issue of low ARM acceptance rate, we propose to apply ARM on an augmented unconstrained MDP, which guides the policy updates towards the feasible action set by a penalty signal induced by the action acceptance events of ARM, instead of on the original action-constrained MDP. Moreover, we show that these two MDPs are equivalent with respect to optimal policies.

**Constructing an augmented MDP.** Based on the original action-constrained MDP $\mathcal{M} = (\mathcal{S}, \mathcal{A}, \mathcal{P}, \gamma, r, \mathcal{C})$, we propose to construct an Augmented Unconstrained Two-Objective MDP (AUTO-MDP) $\tilde{\mathcal{M}} := (\mathcal{S}, \mathcal{A}, \tilde{\mathcal{P}}, \gamma, \tilde{r})$ by adding additional self-loop state transitions and penalty signal for those actions $a \in \mathcal{C}(s)$:

Figure 1: An illustration of AUTO-MDP, where $a \in \mathcal{C}(s)$ and $\tilde{a} \notin \mathcal{C}(s)$.

- The AUTO-MDP $\tilde{\mathcal{M}}$ shares the same state and action spaces with the original MDP $\mathcal{M}$.

- The augmented reward function $\tilde{r} : \mathcal{S} \times \mathcal{A} \to \mathbb{R}^2$ returns a 2-dimensional reward vector $[r(s,a), c(s,a)]$ and is defined as: Let $\mathcal{K} > 0$ be a constant penalty. Then, we construct (i) For any $(s,a)$ with $a \notin \mathcal{C}(s)$, $\tilde{r}(s,a) := [0, -\mathcal{K}]$. (ii) For any $(s,a)$ with $a \in \mathcal{C}(s)$, $\tilde{r}(s,a) := [r(s,a), 0]$.

- The augmented transition kernel $\tilde{\mathcal{P}}$ is defined as follows: For any $(s,a,s')$ with $a \in \mathcal{C}(s)$, let $\tilde{\mathcal{P}}(s'|s,a) = \mathcal{P}(s'|s,a)$. For any $(s,a,s')$ with $a \notin \mathcal{C}(s)$, let

$$\tilde{\mathcal{P}}(s'|s,a) = \begin{cases} 1, & s = s' \\ 0, & \text{otherwise} \end{cases} \tag{2}$$

The idea of AUTO-MDP is illustrated in Figure 1.

Moreover, for any policy $\pi \in \Pi_{\mathcal{A}}$, we define the vector-valued Q function $\boldsymbol{Q} : \mathcal{S} \times \mathcal{A} \to \mathbb{R}^2$ of $\pi$ as $\boldsymbol{Q}(s,a;\pi) := \mathbb{E}\big[\sum_{t=0}^\infty \gamma^t \boldsymbol{r}_t | s_0 = s, a_0 = a; \pi\big]$, which is a natural extension of the standard scalar-valued Q function. Clearly, the vector-valued Q function also satisfies the Bellman equation

$$\boldsymbol{Q}(s,a;\pi) = \boldsymbol{r}(s,a) + \gamma \mathbb{E}_{s' \sim \tilde{\mathcal{P}}, a' \sim \pi}[\boldsymbol{Q}(s',a';\pi)]. \tag{3}$$

As in the standard multi-objective MDPs (MOMDP), the set of optimal policies depend on the preference over the objectives. In the MOMDP literature (Abels et al., 2019; Yang et al., 2019), this is typically characterized by using linear scalarization with a *preference vector* $\boldsymbol{\lambda} = [\lambda_r, \lambda_c]$ such that the scalarized Q value of a policy $\pi$ is defined as $Q_{\boldsymbol{\lambda}}(s,a;\pi) := \langle \boldsymbol{\lambda}, \boldsymbol{Q}(s,a;\pi) \rangle$. Without loss

of generality, we presume that $\boldsymbol{\lambda}$ lies in a two-dimensional probability simplex, i.e., $\lambda_r \geq 0$, $\lambda_c \geq 0$, and $\lambda_r + \lambda_c = 1$.

With such a design, to obtain a policy with a sufficiently high acceptance rate under ARM, the learner shall find a policy that maximizes cumulative reward while minimizing violation penalty.

**Equivalence of AUTO-MDP and original MDP in terms of optimal policies.** We show that the constructed AUTO-MDP and the original MDP are equivalent in the sense that they share the same set of optimal policies. This property can be formally stated in the following proposition.

**Proposition 1** (**Equivalence in optimality**). *Let $\pi^* \in \Pi_{\mathcal{C}}$ be an optimal policy among all the policies in $\Pi_{\mathcal{C}}$ under the original action-constrained MDP $\mathcal{M}$. Then, for any $\boldsymbol{\lambda} \in \Lambda$, the policy $\pi^*$ remains an optimal policy among all the policies in $\Pi_{\mathcal{A}}$ under the AUTO-MDP $\tilde{\mathcal{M}}$.*

The proof of Proposition 1 is provided in Appendix A. This result suggests that solving AUTO-MDP can achieve the same maximum cumulative reward as the original action-constrained MDP while providing incentives for a higher action acceptance rate.

**Remarks on the preference vector $\boldsymbol{\lambda}$.** Given the equivalence property in Proposition 1, to solve the AUTO-MDP, one could select a preference vector $\boldsymbol{\lambda} \in \Lambda$ and find a corresponding optimal policy $\pi^*_{\boldsymbol{\lambda}}$ such that $\boldsymbol{\lambda}^\top \boldsymbol{Q}(s, a; \pi^*_{\boldsymbol{\lambda}}) \geq \boldsymbol{\lambda}^\top \boldsymbol{Q}(s, a; \pi)$, for all $s \in \mathcal{S}, a \in \mathcal{A}$, and $\pi \in \Pi_{\mathcal{A}}$. In practice, different choices of $\boldsymbol{\lambda}$ lead to distinct learning behaviors: (i) If $\lambda_r$ is much larger than $\lambda_c$, then the violation penalty could be too small to enhance the acceptance rate of ARM. (ii) If $\lambda_r$ is much smaller than $\lambda_c$, then high violation penalty could make the policy very conservative and lead to low cumulative reward.

To find a proper $\boldsymbol{\lambda}$, one straightforward approach is to employ hyperparameter tuning over $\boldsymbol{\lambda}$, at the cost of several times more of environment steps. To address this, we propose a practical implementation of ARAM based on multi-objective RL, which can learn well-performing policies for all preferences simultaneously, as described below.

---

**Algorithm 1:** Practical Implementation of ARAM

**Input** : Initial parameters $\phi, \theta$, preference sampling distribution $\rho_{\boldsymbol{\lambda}}$, actor and critic learning rates $\xi_\pi, \xi_Q$

1 Initialize the replay buffer $\mathcal{D}_r$;
2 Initialize the augmented replay buffer $\mathcal{D}_a$;
3 **for** *each iteration $j$* **do**
4     Sample $\boldsymbol{\lambda} \in \Lambda$ according to $\rho_{\boldsymbol{\lambda}}$;
5     **for** *each environment step $t$* **do**
6         Sample $a_t \sim \pi^\dagger_\phi(s_t; \boldsymbol{\lambda})$ by ARM;
7         Obtain augmented reward $\boldsymbol{r}_t = [r_t, c_t]$ and next state $s_{t+1}$;
8         **if** $a_t \in \mathcal{C}(s_t)$ **then**
9             Store $(s_t, a_t, r_t, c_t, s_{t+1})$ in $\mathcal{D}_r$
10         **else**
11             Store $(s_t, a_t, r_t, c_t, s_{t+1})$ in $\mathcal{D}_a$;
12     **for** *each gradient step $\tau$* **do**
13         Draw a mini-batch of samples from $\mathcal{D}_r$ and $\mathcal{D}_a$;
14         Critic update by (4): $\theta \leftarrow \theta - \xi_Q \nabla_{\hat{\theta}} J_Q(\theta)$;
15         Policy update by (5): $\phi \leftarrow \phi - \xi_\pi \nabla_{\hat{\phi}} J_\pi(\phi)$;

---

## 4.3 A Multi-Objective RL Implementation of ARAM

This introduces a practical multi-objective RL implementation of ARAM. As described earlier, the two modifications ARM and AUTO-MDP are general in that they can be employed to adapt any standard deep RL algorithm to ACRL. This work adopts SAC (Haarnoja et al., 2018) as the backbone to showcase how to integrate the proposed modifications into an existing deep RL algorithm.

**Solving AUTO-MDP via multi-objective RL.** To learn policies for all preferences simultaneously, we adapt the multi-objective SAC (MOSAC) presented in (Hung et al., 2023) to the AUTO-MDP:

- **Policy loss and critic loss.** MOSAC also adopts an actor-critic architecture as in vanilla SAC. Let $\phi$ and $\theta$ denote the parameters of the policy and the critic. MOSAC learns a preference-dependent policy $\pi_\phi(a|s; \boldsymbol{\lambda})$ and the corresponding vector-valued Q function by a preference-dependent critic network $\boldsymbol{Q}_\theta(s, a; \boldsymbol{\lambda})$. Given a state-action sampling distri-

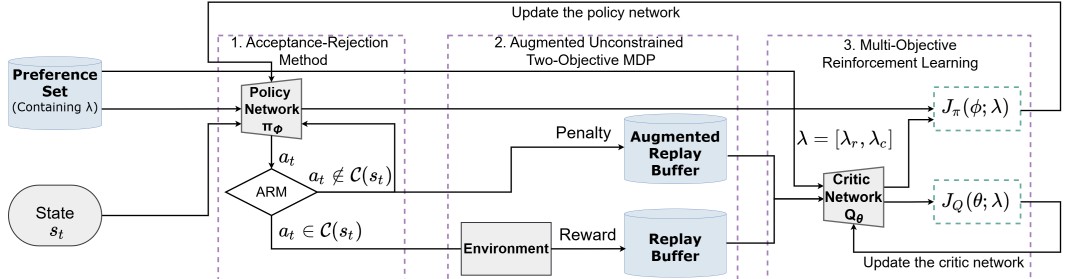

Figure 2: **An illustration of the ARAM framework.** ARAM is composed of three components: (1) ARM: Use an oracle to verify whether the sampled action is in the feasible action set. (2) AUTO-MDP: Assign penalties to invalid actions within an augmented MDP framework, thereby reducing the rate of action violations. (3) MORL: Use MORL to discover well-performing policies under all penalty weights simultaneously.

bution $\mu$, the critic network is updated iteratively by minimizing the following loss function

$$J_{\boldsymbol{Q}}(\theta; \boldsymbol{\lambda}) = \mathbb{E}_{(s,a)\sim\mu}\left[\left(\langle\boldsymbol{\lambda}, \boldsymbol{Q}_\theta\left(s, a; \pi_\phi, \boldsymbol{\lambda}\right) - \left(\boldsymbol{r}\left(s, a\right) + \gamma\mathbb{E}_{s'\sim\mathcal{P}(\cdot|s,a)}\left[\boldsymbol{V}_{\bar{\theta}}\left(s'; \pi_\phi, \boldsymbol{\lambda}\right)\right]\right)\rangle\right)^2\right],$$
(4)

where $\boldsymbol{V}_\theta(s; \pi_\phi, \boldsymbol{\lambda}) := \mathbb{E}_{a\sim\pi_\phi(\cdot|\cdot;\boldsymbol{\lambda})}[\boldsymbol{Q}_\theta(s, a; \pi_\phi, \boldsymbol{\lambda}) - \alpha\log\pi_\phi(a|s; \boldsymbol{\lambda})\mathbf{1}_d]$ with entropy coefficient $\alpha$, $\bar{\theta}$ denotes the parameters of the target critic network. Regarding the policy update, the policy $\pi_\phi$ is updated by minimizing

$$J_\pi(\phi; \boldsymbol{\lambda}) = \mathbb{E}_{s\sim\mu}\left[\mathbb{E}_{a\sim\pi_\phi}\left[\alpha\log\left(\pi_\phi\left(a\mid s; \boldsymbol{\lambda}\right)\right) - \langle\boldsymbol{\lambda}, \boldsymbol{Q}\left(s, a; \pi_\phi, \boldsymbol{\lambda}\right)\rangle\right]\right].$$
(5)

Notably, Equation (4) and Equation (5) can be viewed as the critic loss and the policy loss of vanilla SAC under the scalarized Q function.

- **Dual-buffer design.** Like vanilla SAC, MOSAC is an off-policy algorithm and makes policy and critic updates based on the samples from experience replay buffers. To better address the augmented transitions in AUTO-MDP, we propose a dual-buffer design, where we store the feasible transitions and the augmented infeasible transitions in two separate replay buffers, namely a real replay buffer $\mathcal{D}_r$ and an augmented replay buffer $\mathcal{D}_a$. This design offers more flexibility in balancing the number of updates by feasible and infeasible transitions, especially at the initial training stage when the action violation rate is high.

- **Preference distribution.** To update the policy and the critic for different preferences, the preference $\boldsymbol{\lambda}$ is drawn from some distribution $\rho_{\boldsymbol{\lambda}}$. One natural choice is to set $\rho_{\boldsymbol{\lambda}}$ as a uniform distribution over the two-dimensional probability simplex, or essentially a Dirichlet distribution with concentration parameter equal to 1.

The training process is illustrated in Figure 2, with the pseudo code provided in Algorithm 1.

## 5 EXPERIMENTS

**Benchmark Methods.** We compare ARAM with various recent benchmark ACRL algorithms, including NFWPO, DPre+, SPre+, and FlowPG. NFWPO (Lin et al., 2021) achieves favorable constraint satisfaction at the cost of high QP overhead as it enforces action constraints by Frank-Wolfe search. DPre+ and SPre+, proposed by (Kasaura et al., 2023), adapt the vanilla DDPG (Lillicrap et al., 2016) and SAC (Haarnoja et al., 2018) to ACRL by using a QP-based projection step for action post-processing, learning the critic with pre-projected actions, and applying a penalty term to guide the policy updates. For a fair comparison, we use the official implementation and the hyperparameter settings of DPre+, SPre+, and NFWPO provided by (Kasaura et al., 2023). FlowPG enforces action constraints via a pre-trained Normalizing Flow model, and we use the official source code provided by (Brahmanage et al., 2023). For the testing of ARAM, we set $\boldsymbol{\lambda} = [0.9, 0.1]$ as the default input preference of the policy network $\pi_\phi(\cdot|\boldsymbol{\lambda})$. Moreover, during the testing of all the above algorithms, an auxiliary projection step is employed to guarantee that actions used for environment interaction always satisfy the action constraints.

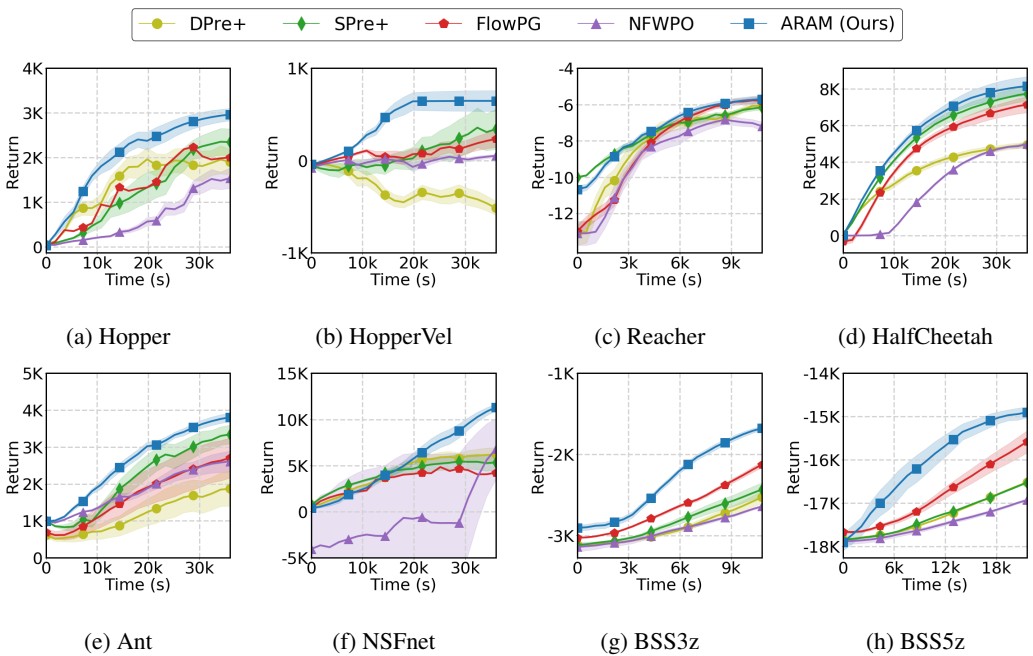

Figure 3: Learning curves of ARAM and the benchmark methods across different environments in terms of the evaluation return.

**Evaluation Domains.** We evaluate the algorithms in various benchmark domains widely used in the ACRL literature (Lin et al., 2021; Kasaura et al., 2023; Brahmanage et al., 2023): (i) *MuJoCo locomotion tasks* (Todorov et al., 2012): These tasks involve training robots to achieve specified goals, such as running forward and controlling their speed within certain limits. (ii) *Resource allocation for networked systems*: These tasks involve properly allocating resource under capacity constraints, including NSFnet and Bike Sharing System (BSS) (Ghosh & Varakantham, 2017). For NSFnet, the learner needs to allocate packets of different flows to multiple communication links. The action constraints are induced by the per-link maximum total assigned packet arrival rate. We follow the configuration provided by (Lin et al., 2021) and use the open-source network simulator from PCC-RL (Jay et al., 2019). For BSS, the environment consists of $m$ bikes and $n$ stations, each with a capacity limit of $c$. The learner needs to reallocate bikes to different stations based on the current situation. We follow the experimental scenario of (Lin et al., 2021) and evaluate our approach on two tasks: BSS3z with $n = 3$ and $m = 90$ and BSS5z with $n = 5$ and $m = 150$. Their capacities are both set to 40. A detailed description about these tasks is provided in Appendix D.1.

**Performance Metrics.** We evaluate the performance in the following aspects:

- *Training efficiency*: We record the evaluation returns at different training stages, in terms of both the wall clock time and the environment steps. To ensure fair measurements of wall clock time, we run each algorithm independently using the same computing device. Moreover, we report the cumulative number of QP operations as an indicator of the training computational overhead.
- *Valid action rate*: At the testing phase, we evaluate the valid action rate by sampling 100 actions from the policy network at each step of an episode. This metric reflects how effectively each method enforces the action constraints throughout the evaluation phase.
- *Per-action inference time*: The efficiency of action inference reflects the design complexity, such as the need for generative models and QP operations, of each ACRL method. During evaluation, we measure the per-action inference time for 1 million actions. This inference time serves as a critical metric for ACRL deployment.

## 5.1 EXPERIMENTAL RESULTS

Our proposed method effectively reduces the use of costly QP operations, allowing us to address ACRL with a more lightweight framework. This subsection demonstrates the effectiveness of our

method in training efficiency, valid action rate, and per-action inference time. Unless stated otherwise, all the results reported below are averaged over five random seeds.

**Does the proposed method outperform other ACRL benchmark methods in cumulative rewards?** Figure 3 shows the evaluation reward versus wall clock time, and Table 2 and Figure 8 (in Appendix C) show the valid action rates. We observe that ARAM enjoys fast learning progress and high valid action rates simultaneously. On the other hand, due to the reliance on the projection layer, the projection-based methods like DPre+ and SPre+ have relatively very low valid action rates, which require a large number of QP operations and hence lead to a longer training time. Additionally, they perform poorly in resource allocation environments since the optimal solutions for these problems cannot be directly found through projection. Regarding NFWPO and FlowPG, these two methods can both effectively learn policies that satisfy the constraints but not necessarily achieve a high average return. Moreover, the results regarding sample efficiency, measured by the evaluation return versus environment steps, can be found in Appendix C.

Table 2: Comparison of ARAM and the benchmark algorithms in terms of the valid action rate.

| Environment | DPre+ | SPre+ | NFWPO | FlowPG | ARAM (Ours) |
|---|---|---|---|---|---|
| Hopper | **0.97 ± 0.03** | 0.40 ± 0.01 | **0.99 ± 0.01** | **0.98 ± 0.01** | **0.99 ± 0.01** |
| HopperVel | 0.38 ± 0.30 | 0.60 ± 0.09 | **0.94 ± 0.03** | 0.71 ± 0.19 | 0.80 ± 0.09 |
| Reacher | 0.25 ± 0.16 | 0.93 ± 0.03 | **0.97 ± 0.02** | **0.95 ± 0.01** | **0.98 ± 0.01** |
| HalfCheetah | 0.35 ± 0.06 | 0.78 ± 0.19 | **0.97 ± 0.01** | 0.78 ± 0.17 | 0.78 ± 0.29 |
| Ant | 0.41 ± 0.38 | 0.29 ± 0.07 | **0.99 ± 0.01** | 0.83 ± 0.17 | **1.00 ± 0.00** |
| NSFnet | 0.00 ± 0.00 | 0.04 ± 0.01 | 0.92 ± 0.05 | **0.98 ± 0.01** | **0.94 ± 0.04** |
| BSS3z | 0.25 ± 0.02 | 0.28 ± 0.24 | **0.73 ± 0.18** | 0.59 ± 0.20 | **0.72 ± 0.24** |
| BSS5z | 0.13 ± 0.11 | 0.31 ± 0.16 | **0.84 ± 0.14** | 0.68 ± 0.17 | 0.77 ± 0.16 |

**Does our proposed method achieve a lower training and inference overhead?** QP operations are known to be computationally costly and thereby account for a substantial fraction of training time in many ACRL methods. Figure 4 presents the log-scale plot of QP usage under different algorithms. It is evident that ARAM exhibits significantly lower QP computation compared to others. Due to its reduced dependency on QP operations, ARAM achieves higher training efficiency.

Moreover, Table 3 shows the average per-action inference time during evaluation. ARAM benefits from computationally efficient action inference due to its almost QP-free design, as only a minimal subset of policy output actions that violate constraints requires the QP operator. By contrast, the projection-based methods like DPre+ and SPre+ and the flow-based FlowPG all suffer from much higher per-action inference time.

Table 3: Comparison of ARAM and the benchmark algorithms in terms of the evaluation time. The evaluation times are calculated through one million evaluation steps.

| Environment | DPre+ (s) | SPre+ (s) | NFWPO (s) | FlowPG (s) | ARAM (Ours) (s) |
|---|---|---|---|---|---|
| Reacher | 71.47 ± 5.63 | 303.01 ± 60.81 | 67.43 ± 5.02 | 71.47 ± 3.12 | **44.72 ± 0.93** |
| HalfCheetah | 211.67 ± 48.70 | 253.53 ± 30.48 | 180.41 ± 10.57 | 210.63 ± 7.51 | **57.14 ± 8.13** |
| Hopper | 99.43 ± 6.71 | 175.43 ± 9.47 | 88.71 ± 11.41 | 103.12 ± 5.13 | **61.72 ± 2.15** |
| HopperVel | 177.31 ± 15.73 | 97.12 ± 7.95 | 82.63 ± 7.13 | 91.43 ± 5.17 | **63.81 ± 10.04** |

**Comparison: ARAM versus RL for constrained MDPs with projection.** To make the comparison even more comprehensive, we also adapt FOCOPS (Zhang et al., 2020), which is a popular RL approach designed for constrained MDPs to address long-term discounted cost, to ACRL by adding a projection step. Table 4 and Table 5 present a comparison of ARAM and FOCOPS in the evaluation rewards and the valid action rates. These results suggest that ACRL requires fundamentally different solutions from RL for constrained MDPs.

**Ablation study on MORL.** To investigate the benefits of using MORL, we perform an ablation study that compares the MORL implementation with a single-objective variant with a fixed preference (termed SOSAC below). From Figure 5, MORL can discover policies with both competitive final forward reward and favorable constraint satisfaction due to the implicit knowledge sharing across preferences during training. By contrast, SOSAC under a fixed preference fails to meet both criteria simultaneously, and this suggests that direct hyperparameter tuning can be rather ineffective. We also provide the learning curves in Figure 11 in Appendix C.

## 6 CONCLUSION

In this paper, we introduced ARAM, a novel framework designed to address ACRL problems by augmenting standard deep RL algorithms. By employing the acceptance-rejection method and an

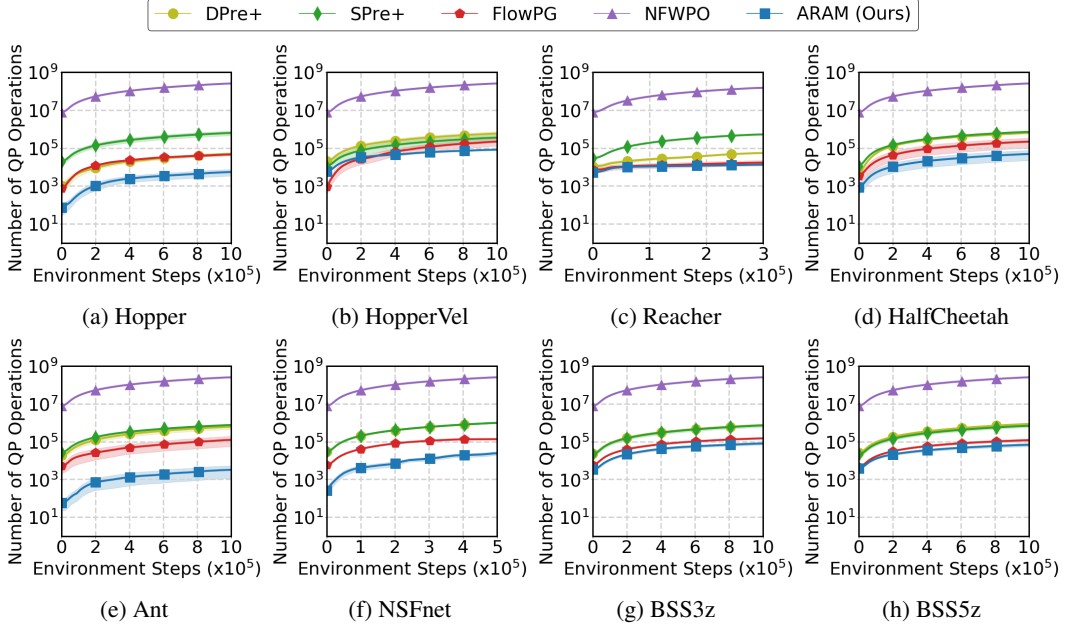

Figure 4: Cumulative number of QP operations of ARAM and the benchmark methods across different environments, with the y-axis on a logarithmic scale.

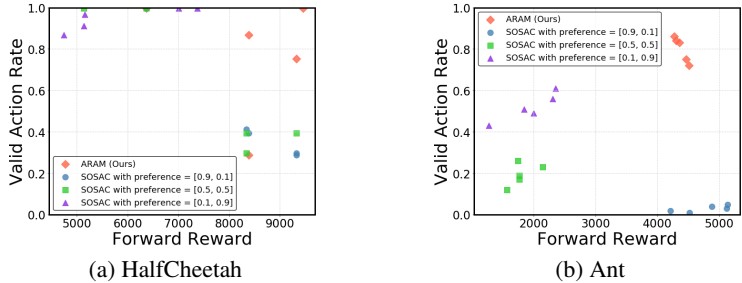

Figure 5: **Ablation study of MORL.** We plot the tuples of forward reward and valid action rate of ARAM and the three single-objective SAC variants with fixed preferences $\boldsymbol{\lambda} = [0.9, 0.1], [0.5, 0.5]$, and $[0.1, 0.9]$ in HalfCheetah and Ant. The five markers of each color refer to the results of the same algorithm over five distinct random seeds.

Table 4: Comparison of ARAM and FO-COPS in terms of the evaluation return.

| Environment | FOCOPS | ARAM (Ours) |
|---|---|---|
| Hopper ($\times 10^3$) | $2.27 \pm 0.41$ | $\mathbf{3.07 \pm 0.24}$ |
| HopperVel ($\times 10^2$) | $0.07 \pm 3.13$ | $\mathbf{6.49 \pm 2.31}$ |
| Reacher ($\times 10^0$) | $-5.80 \pm 1.14$ | $\mathbf{-4.78 \pm 0.33}$ |
| HalfCheetah ($\times 10^3$) | $6.08 \pm 1.87$ | $\mathbf{8.38 \pm 1.11}$ |
| Ant ($\times 10^3$) | $3.06 \pm 1.11$ | $\mathbf{5.00 \pm 0.32}$ |
| NSFnet ($\times 10^4$) | $0.68 \pm 0.16$ | $\mathbf{1.32 \pm 0.08}$ |
| BSS3z ($\times 10^3$) | $-1.93 \pm 0.20$ | $\mathbf{-1.65 \pm 0.04}$ |
| BSS5z ($\times 10^4$) | $-1.61 \pm 0.05$ | $\mathbf{-1.51 \pm 0.02}$ |

Table 5: Comparison of ARAM and FO-COPS in terms of the valid action rate.

| Environment | FOCOPS | ARAM (Ours) |
|---|---|---|
| Hopper | $0.22 \pm 0.15$ | $\mathbf{0.99 \pm 0.01}$ |
| HopperVel | $0.32 \pm 0.11$ | $\mathbf{0.80 \pm 0.09}$ |
| Reacher | $0.87 \pm 0.10$ | $\mathbf{0.98 \pm 0.01}$ |
| HalfCheetah | $0.59 \pm 0.13$ | $\mathbf{0.78 \pm 0.29}$ |
| Ant | $0.26 \pm 0.09$ | $\mathbf{1.00 \pm 0.00}$ |
| NSFnet | $0.46 \pm 0.28$ | $\mathbf{0.94 \pm 0.04}$ |
| BSS3z | $0.45 \pm 0.19$ | $\mathbf{0.72 \pm 0.24}$ |
| BSS5z | $0.16 \pm 0.13$ | $\mathbf{0.77 \pm 0.16}$ |

augmented MDP, ARAM effectively reduces the need for costly QP operations and improves valid action rates. Our experimental results demonstrate that ARAM can simultaneously achieve faster learning progress and require significantly fewer QP operations than the existing ACRL methods.

ACKNOWLEDGMENTS

This material is based upon work partially supported by National Science and Technology Council (NSTC), Taiwan under Contract No. NSTC 113-2628-E-A49-026, and the Higher Education Sprout Project of the National Yang Ming Chiao Tung University and Ministry of Education, Taiwan. Shao-Hua Sun was supported by the Yushan Fellow Program by the Ministry of Education, Taiwan. We also thank the National Center for High-performance Computing (NCHC) for providing computational and storage resources.

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

APPENDICES

# Table of Contents

## A  PROOF OF PROPOSITION 1

For ease of exposition, we restate the proposition as follows.

**Proposition 1 (Equivalence in optimality).** *Let $\pi^* \in \Pi_{\mathcal{C}}$ be an optimal policy among all the policies in $\Pi_{\mathcal{C}}$ under the original action-constrained MDP $\mathcal{M}$. Then, for any $\boldsymbol{\lambda} \in \Lambda$, the policy $\pi^*$ remains an optimal policy among all the policies in $\Pi_{\mathcal{A}}$ under the AUTO-MDP $\tilde{\mathcal{M}}$.*

*Proof.* Let us first fix a preference vector $\boldsymbol{\lambda} \in \Lambda$. Recall that $\boldsymbol{\lambda} = [\lambda_r, \lambda_c]$. Let $\pi^*_{\boldsymbol{\lambda}} \in \Pi_{\mathcal{A}}$ be a deterministic optimal policy of the AUTO-MDP under the preference vector $\boldsymbol{\lambda}$. By the classic literature of MDPs, we know such a $\pi^*_{\boldsymbol{\lambda}}$ must exist. Then, we have

$$\langle \boldsymbol{\lambda}, \boldsymbol{Q}(s, a; \pi^*_{\boldsymbol{\lambda}}) \rangle \geq \langle \boldsymbol{\lambda}, \boldsymbol{Q}(s, a; \pi) \rangle, \tag{6}$$

for all $s \in \mathcal{S}$, for all $a \in \mathcal{A}$, and for all $\pi \in \Pi_{\mathcal{A}}$. In the sequel, we slightly abuse the notation and use $\pi(s)$ to denote the deterministic action taken at state $s$ by a deterministic policy $\pi$.

To prove the proposition, we just need to show that such a $\pi^*_{\boldsymbol{\lambda}}$ must also be in the set of feasible policies $\Pi_{\mathcal{C}}$. We prove this by contradiction. Suppose $\pi^*_{\boldsymbol{\lambda}} \notin \Pi_{\mathcal{C}}$. Then, there must exist a state $\bar{s} \in \mathcal{S}$ such that $\pi^*_{\boldsymbol{\lambda}}(\bar{s}) \notin \mathcal{C}(s)$. Then, we know

$$\langle \boldsymbol{\lambda}, \boldsymbol{Q}(\bar{s}, \pi^*_{\boldsymbol{\lambda}}(\bar{s}); \pi^*_{\boldsymbol{\lambda}}) \rangle = \langle \boldsymbol{\lambda}, [0, -\mathcal{K}]^\top + \gamma \boldsymbol{Q}(\bar{s}, \pi^*_{\boldsymbol{\lambda}}(\bar{s}); \pi^*_{\boldsymbol{\lambda}}) \rangle \tag{7}$$

$$= \lambda_c \cdot \frac{-\mathcal{K}}{1 - \gamma}, \tag{8}$$

where Equation (7) follows from the Bellman equation and the self-loop transitions of AUTO-MDP, and Equation (8) holds by recursively rolling out the self-loop transitions. Let us pick a feasible action $a \in \mathcal{C}(s)$. We know such an action $a$ must exist as $\mathcal{C}(s)$ is assumed non-empty. Then, we

have

$$\langle \boldsymbol{\lambda}, \boldsymbol{Q}(\bar{s}, a; \pi_{\boldsymbol{\lambda}}^*) \rangle = \langle \boldsymbol{\lambda}, [r(\bar{s}, a), 0]^\top + \gamma \mathbb{E}_{s' \sim \tilde{\mathcal{P}}(\cdot | \bar{s}, a)} [\boldsymbol{Q}(s', \pi_{\boldsymbol{\lambda}}^*(s'); \pi_{\boldsymbol{\lambda}}^*)] \rangle \tag{9}$$

$$\geq \lambda_r \cdot r(\bar{s}, a) + \gamma \lambda_c \cdot \frac{-\mathcal{K}}{1 - \gamma} \tag{10}$$

$$> \lambda_c \cdot \frac{-\mathcal{K}}{1 - \gamma}, \tag{11}$$

where Equation (9) follows from the Bellman equation, Equation (10) holds by considering the worst-case reward and penalty, and Equation (11) holds due to the non-negativity of the reward function and that $\gamma < 1$. Therefore, by combining Equation (8) and Equation (11), we know $\boldsymbol{Q}_{\boldsymbol{\lambda}}(\bar{s}, a; \pi_{\boldsymbol{\lambda}}^*) \equiv \langle \boldsymbol{\lambda}, \boldsymbol{Q}(\bar{s}, a; \pi_{\boldsymbol{\lambda}}^*) \rangle > \langle \boldsymbol{\lambda}, \boldsymbol{Q}(\bar{s}, \pi_{\boldsymbol{\lambda}}^*(\bar{s}); \pi_{\boldsymbol{\lambda}}^*) \rangle \equiv \boldsymbol{Q}_{\boldsymbol{\lambda}}(\bar{s}, \pi_{\boldsymbol{\lambda}}^*(\bar{s}); \pi_{\boldsymbol{\lambda}}^*)$. Then, by the argument of the standard one-step greedy policy improvement, we know $\pi_{\boldsymbol{\lambda}}^*$ can be improved by taking action $a$ at state $\bar{s}$ instead. Hence, $\pi_{\boldsymbol{\lambda}}^*$ cannot be an optimal policy. This completes the proof. $\square$

## B   DETAILED ARCHITECTURE OF ARAM

Our implementation is based on Q-Pensieve (Hung et al., 2023), Figure 6 illustrates the complete training process of ARAM. MORL aims to optimize policies under varying preferences, which poses challenges in sample efficiency. To address this, Q-Pensieve introduces a memory-sharing technique via the Q Replay Buffer, which stores past critic-network snapshots. This approach enables leveraging past experiences effectively, improving sample efficiency by allowing the agent to revisit and learn from previous interactions across different preference settings. By incorporating the Q Replay Buffer, ARAM facilitates efficient policy learning, allowing the agent to balance performance improvement and constraint satisfaction.

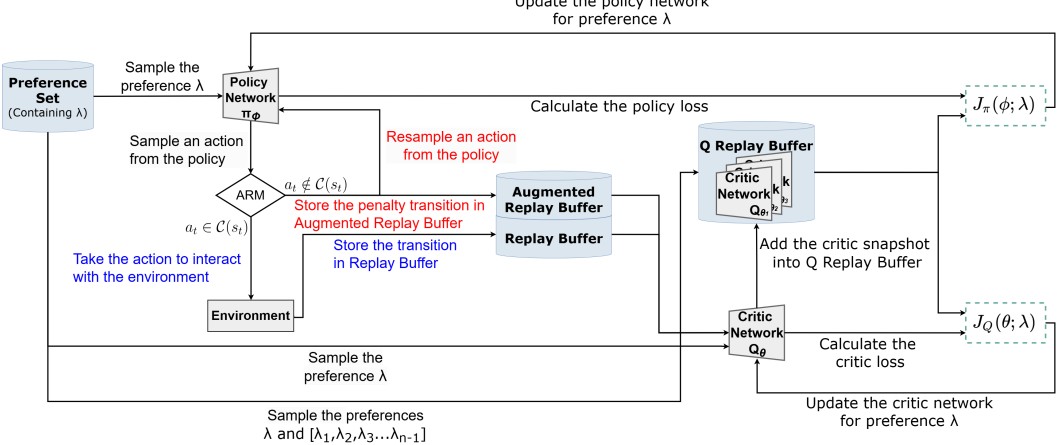

Figure 6: **Overview of the training process of ARAM.** When an action is sampled from the policy, if it belongs to the feasible set, the transition is stored in the Replay Buffer (blue). Otherwise, the corresponding penalty transition is stored in the Augmented Replay Buffer, and another action is resampled (red). These stored transitions are then used to update the policy and critic networks.

# C   ADDITIONAL EXPERIMENTAL RESULTS

In this section, we compare ARAM with other baselines, including action-constrained RL and RL for Constrained MDPs. The learning curves demonstrate that ARAM achieves better sample efficiency. Additionally, we conduct an ablation study on the MORL framework, investigating how varying preferences within MORL impact the performance of ARAM.

## C.1   COMPARISON: ARAM VERSUS THE BENCHMARK ALGORITHMS IN TERMS OF SAMPLE EFFICIENCY

**Does ARAM have better sample efficiency?** Figure 7 shows the evaluation rewards, and Figure 8 displays the action acceptance rates. We can observe that ARAM achieves consistently the best reward performance while maintaining a low action violation rate. On the other hand, FlowPG and NFWPO can effectively control the actions within the feasible set, but their reward performance appears lower. In contrast, QP-based methods exhibit a significantly higher rate of action violations.

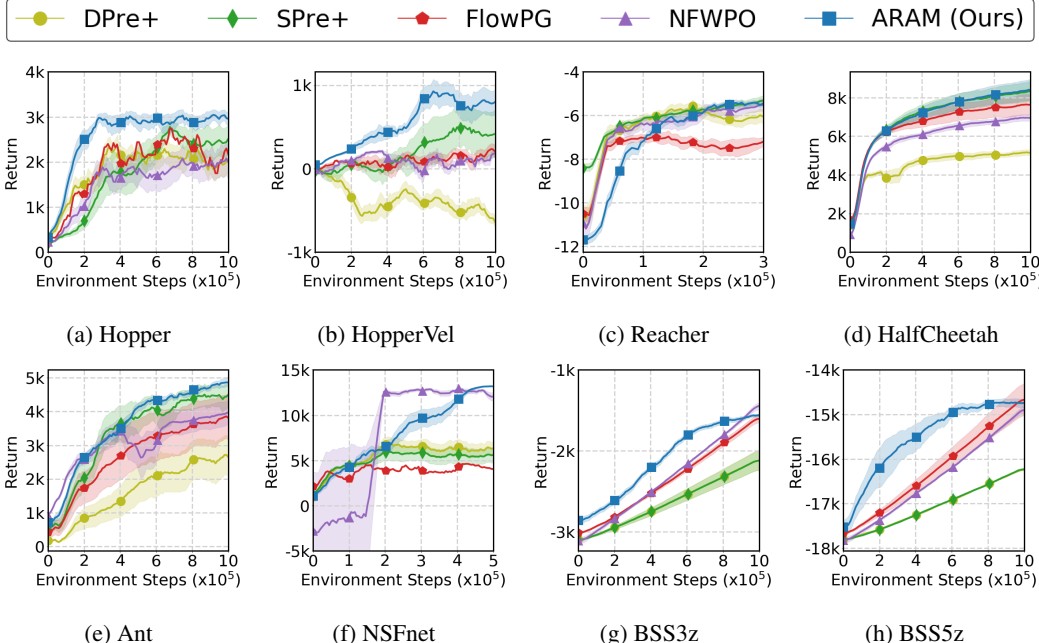

Figure 7: Learning curves of ARAM and the benchmark algorithms across different environments in terms of the evaluation return.

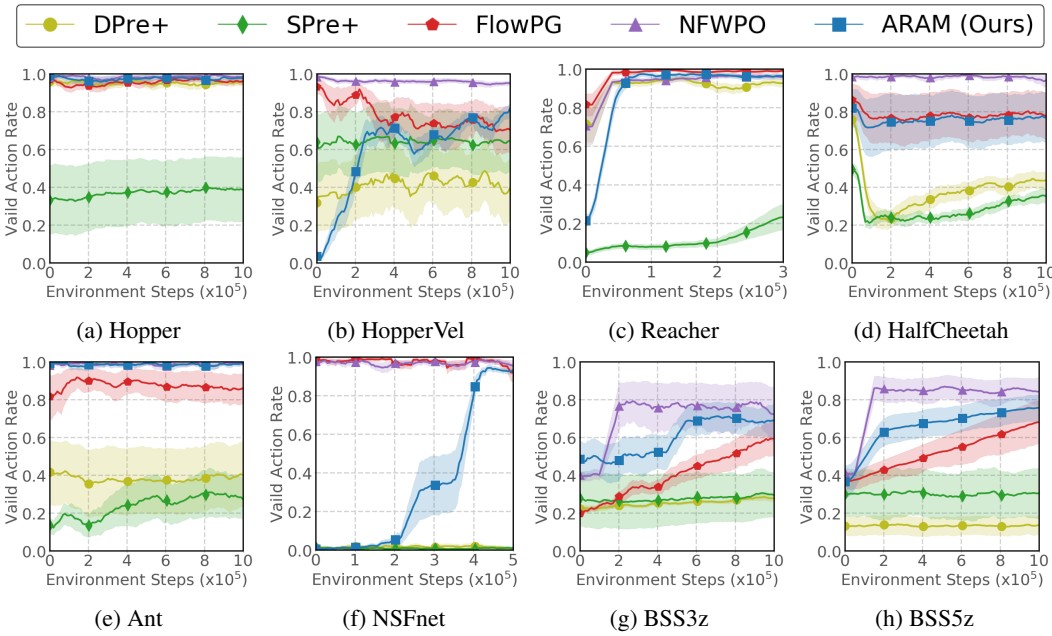

Figure 8: Learning curves of ARAM and the benchmark algorithms across different environments in terms of the valid action rate.

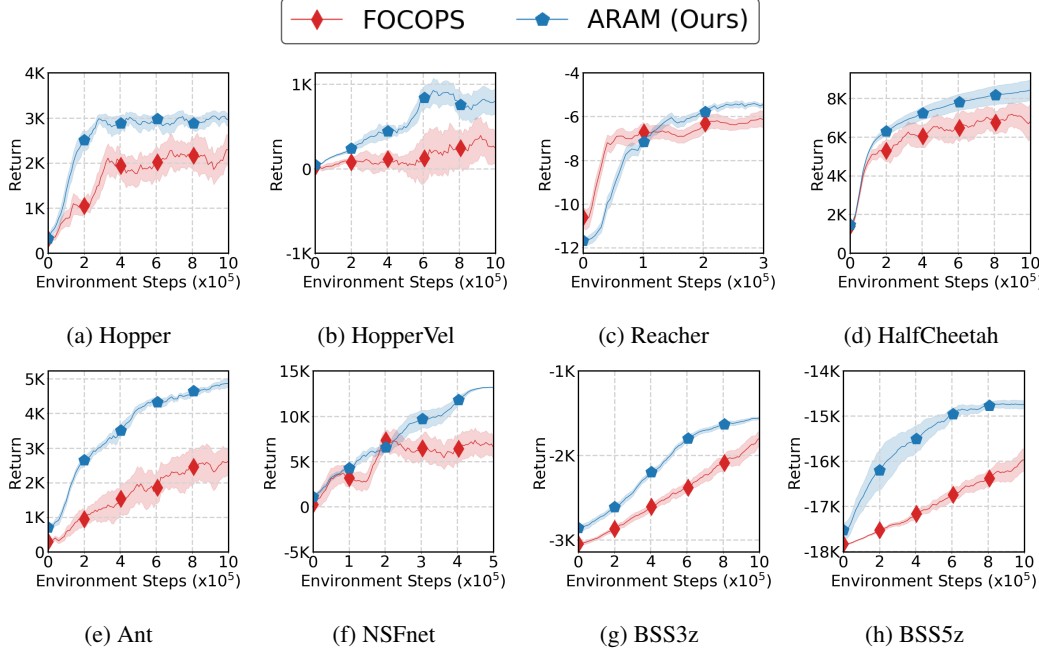

Figure 9: Comparison of the evaluation return of ARAM and FOCOPS across various environments.

## C.2  COMPARISON: ARAM VERSUS RL FOR CONSTRAINED MDPS WITH PROJECTION.

**Does ARAM more effectively reduce the number of action violations while achieving better performance?** Figure 9 and Figure 10 present the training curves of total return and valid action rate, respectively. From these curves, we observe that ARAM consistently outperforms FOCOPS in both the evaluation return and the valid action rate, indicating its superior ability to handle action constraints while optimizing performance.

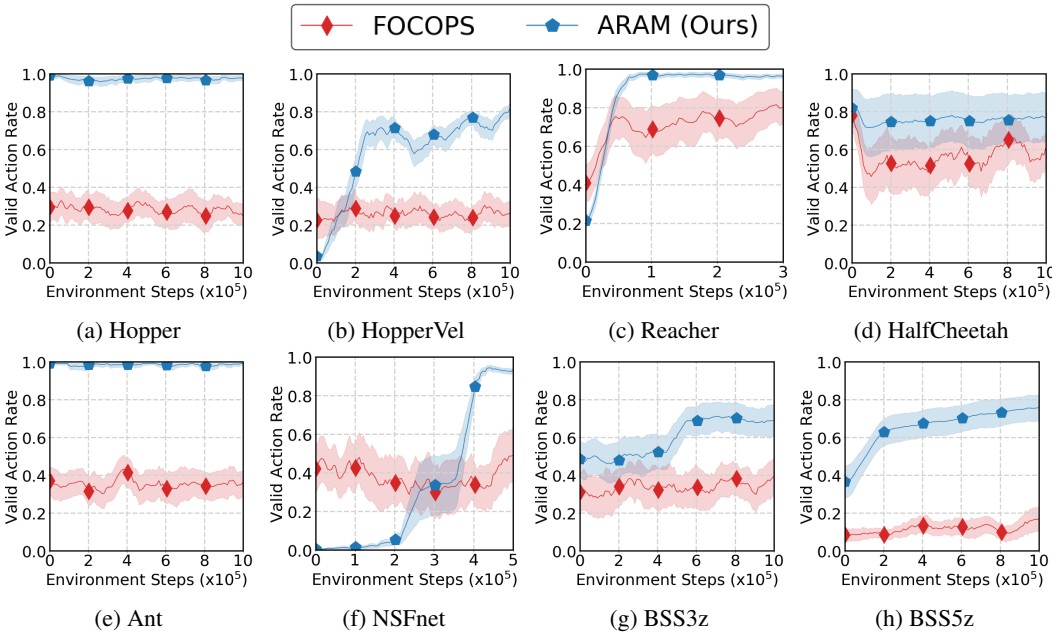

Figure 10: Comparison of the valid action rate of ARAM and FOCOPS across various environments.

### C.3 ABLATION STUDY: THE LEARNING CURVES OF ARAM VERSUS SOSAC

We present the learning curves of the total return and the valid action rate in Figure 11. We can observe that the ARAM consistently achieves higher returns compared to the SOSAC variant with fixed preferences.

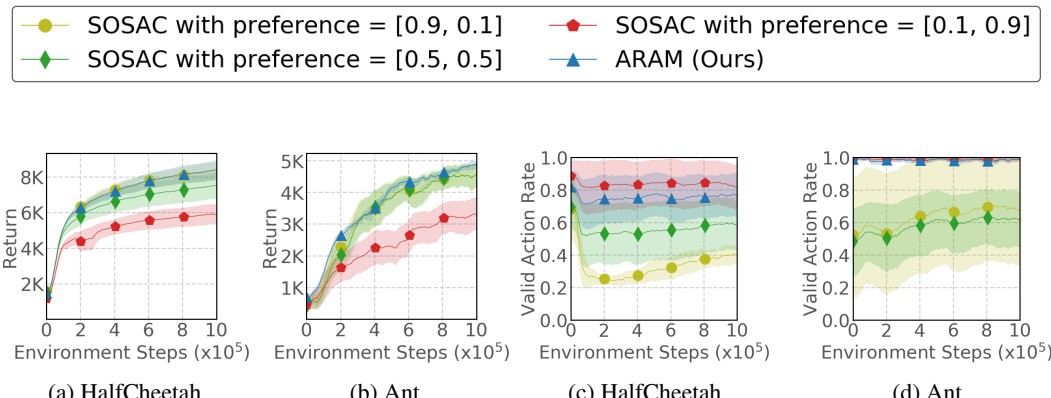

Figure 11: **Ablation Study on MORL.** These figures present the learning curves of ARAM and three single-objective SAC variants with different fixed preferences in terms of the evaluation return (Figures 11a and 11b) and the valid action rate (Figures 11c and 11d) in the HopperVel and Ant environments.

### C.4 SENSITIVITY ANALYSIS: EFFECT OF VIOLATION PENALTY IN ARAM

We present the results of a hyperparameter sensitivity study on the violation penalty $\mathcal{K}$ in the Hopper-Vel and Ant environments, as shown in Figure 12. This study examines how different fixed values

of $\mathcal{K}$ impact the return and valid action rate during training. The findings demonstrate that ARAM maintains stable performance across varying $\mathcal{K}$ values, indicating that the model is largely insensitive to this hyperparameter. This supports our original experimental setup, where a single $\mathcal{K}$ value was used consistently across all tasks and environments without requiring task-specific tuning.

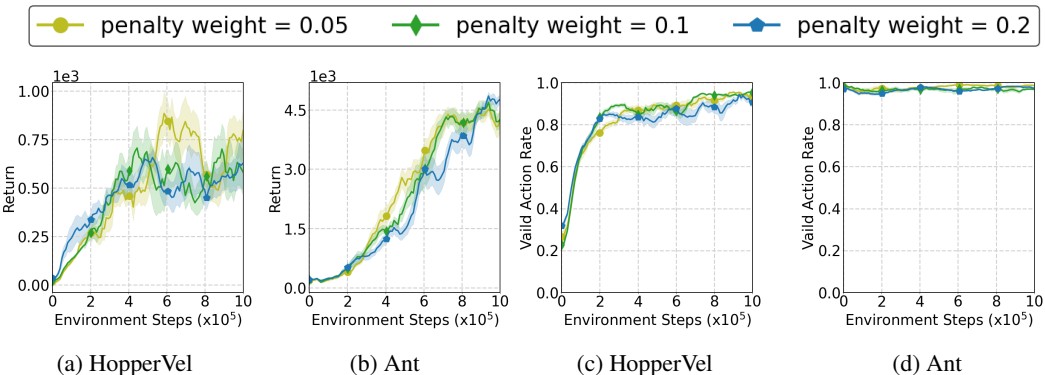

(a) HopperVel       (b) Ant       (c) HopperVel       (d) Ant

Figure 12: **Sensitivity Analysis of the Violation Penalty** $\mathcal{K}$. These figures provide a comparison of the return and the valid action rate across training steps for different values of $\mathcal{K}$ ($\mathcal{K} = 0.05, 0.1, 0.2$) in the HopperVel and Ant environments. These figures show that ARAM is not highly sensitive to the penalty weight.

### C.5 Sensitivity Analysis: Effect of Target Distribution in ARAM

We analyze the impact of different target distributions in ARAM, specifically examining whether this choice significantly affects the final performance. Since ARAM is implemented based on the MOSAC framework, the policy network outputs the mean and standard deviation of a Gaussian distribution, denoted as: $\pi_\phi(a|s) \sim \mathcal{N}(\mu, \sigma^2)$ Following the notation in the paper, we configure an alternative (constrained) target distribution, $\pi^\dagger(a|s)$, by leveraging the (unconstrained) Student-t distribution: $\tilde{\pi}_\phi(a|s) \sim T_\nu(\mu, \sigma^2)$ where $T_\nu$ denotes a Student-t distribution with degrees of freedom $\nu$, mean $\mu$, and variance $\sigma^2$. The constrained target distribution is then defined as:

- $\pi^\dagger_\phi(a|s) \propto \tilde{\pi}_\phi(a|s)$ for $a \in \mathcal{C}(s)$.
- $\pi^\dagger_\phi(a|s) = 0$ for $a \notin \mathcal{C}(s)$.

Notably, the Gaussian distribution is a special case of the Student-t distribution with $\nu = \infty$. The Student-t distribution has heavier tails compared to the Gaussian, potentially leading to a more exploratory action distribution. For ARM, the scaling factor $M$ can be set as: $M = M_0 / \left( \int_{a \in \mathcal{C}(s)} \tilde{\pi}_\phi(a|s) da \right)$, where $M_0$ is a constant dependent on $\nu$. The return and valid action rate is shown on Figure 13. We observe that ARAM can have a slight performance improvement under this new target distribution (especially at the early training stage) and is not very sensitive to this choice. This also corroborates the nice flexibility in the choice of target distribution.

### C.6 Additional Experiment: Scalability of ARAM

To demonstrate the scalability of ARAM to higher-dimensional environments, we extend the NSFnet environment to a 20-dimensional action space called NSFnet20d. Specifically, the agent manages packet delivery across a classic T3 NSFNET network backbone by simultaneously handling 20 packet flows, each with distinct routing paths. The action is defined as the rate allocation of each flow along each candidate path. In total, there are 30 communication links shared by these 20 different flows. The action constraints ensure that the distribution of packets on these shared links stays within the bandwidth capacity, defined as 50 units for each link. Figure 14 presents the results, illustrating ARAM's superior performance in the evaluation return, the valid action rate, and the QP efficiency compared to ACRL benchmark methods. ARAM consistently achieves a higher action

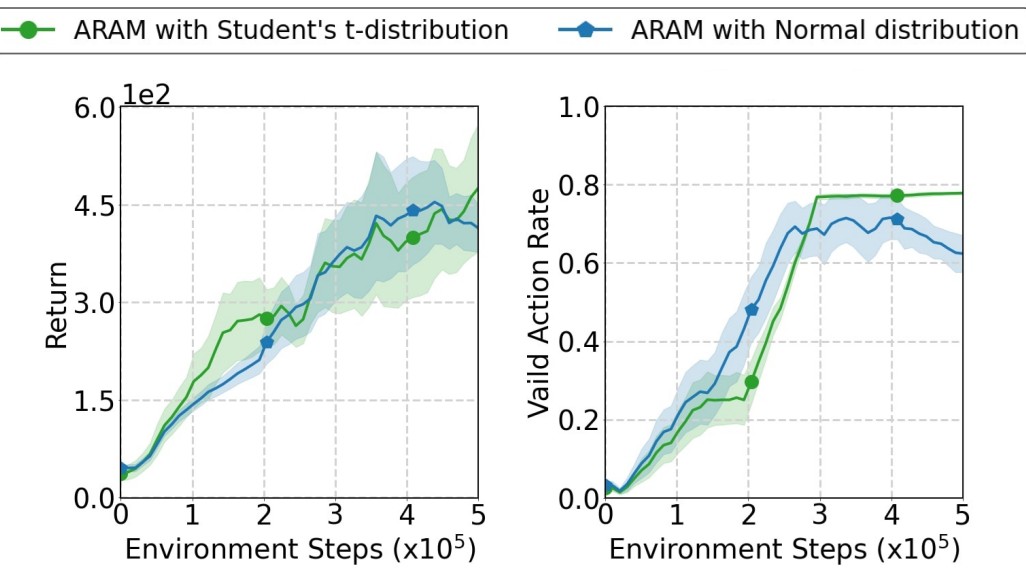

Figure 13: **Sensitivity Analysis of the Target Distribution in ARAM.** These figures provide a comparison of the return and the valid action rate across training steps for different target distributions: Student's t-distribution and the Normal distribution. These figures show that ARAM is not highly sensitive to the choice of target distribution.

validity rate while maintaining robust learning under high-dimensional action spaces and complex constraints, demonstrating its adaptability and efficiency.

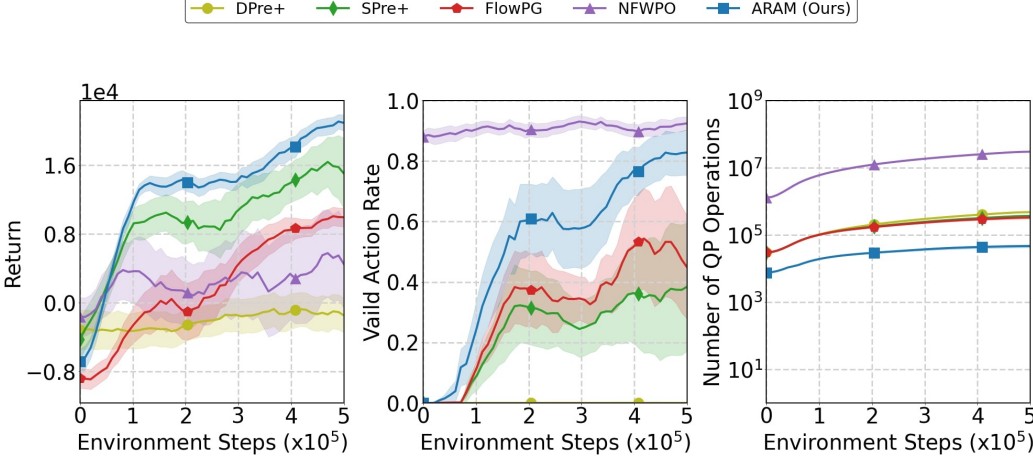

Figure 14: Comparison of the evaluation return, the valid action rate, and the number of QP operations in the NSFnet20d environment.

# D    DETAILED EXPERIMENTAL CONFIGURATIONS

## D.1    ENVIRONMENTS AND ACTION CONSTRAINTS

In this section, we provide the details about the experimental setups and the action constraints considered in different environments.

- MuJoCo: The action is defined as a vector $(a_1, a_2, \cdots, a_k)$, with each element representing the torque applied to a specific joint, and $(w_1, w_2, \cdots, w_k)$ denotes the angular velocities of the joints. The tasks considered in our experiments include:

  - **Reacher, HalfCheetah, Hopper, and Ant**: We use the experimental settings and objectives provided by OpenAI Gym V3 to control the agents in these environments. Each environment presents unique challenges and objectives, such as reaching a target location, maintaining balance, or achieving high-speed movement.
  - **HopperVel**: This task is specifically designed to control the robot to maintain a target horizontal velocity of 3 m/s, differing from the standard Hopper's goal of forward hopping.

- NSFnet: Based on the T3 NSFNET Backbone as discussed in (Lin et al., 2021), this network consists of 9 different packet flows, each with distinct routing paths. There are 8 communication links shared by different flows. The action is defined as the rate allocation of each flow along each candidate path. The action constraint is to check if the distribution of packets on these shared links stays within the bandwidth limits, defined as 50 units for each link. To define action constraints, we described the eight-tuple $(\text{link}_1, ..., \text{link}_8)$, each containing the total amount of flow that pass through that specific link.

- BSS: The action is to allocate bikes to stations under random demands.

  - **BSS3z**: There are 3 stations with a total of 90 bikes ($m = 90$, $n = 3$), and each station has a capacity of 40 bikes.
  - **BSS5z**: This system comprises 5 stations with a total of 150 bikes ($m = 150$, $n = 5$), and each station also has a capacity of 40 bikes.

Table 6: This table provides an overview of the action constraints applied in each experiment environment.

| Environment | Action Constraint |
|---|---|
| HopperVel | $\sum_{i=1}^{3} \max(w_i a_i, 0) \leq 10$ |
| Hopper | $\sum_{i=1}^{3} \max(w_i a_i, 0) \leq 10$ |
| Reacher | $a_1^2 + a_2^2 \leq 0.05$ |
| HalfCheetah | $\sum_{i=1}^{6} |w_i a_i| \leq 20$ |
| Ant | $\sum_{i=1}^{8} a_i^2 \leq 2$ |
| NSFnet | $\sum_{i \in link_j} a_i \leq 50, \quad \forall j \in \{1, 2, \ldots, 8\}$ |
| BSS3z | $\left| \sum_{i=1}^{3} a_i - 90 \right| \leq 5, a_i <= 40$ |
| BSS5z | $\left| \sum_{i=1}^{5} a_i - 150 \right| \leq 5, a_i <= 40$ |

## D.2    HYPERPARAMETERS OF ARAM

We conduct all the experiments with the following hyperparameters.

# E    DETAILED INTRODUCTION TO THE EXISTING ACRL METHODS

## E.1    ACTION PROJECTION

An action produced by the neural network is not guaranteed to remain within the feasible action set. To address this, there are various frameworks to map the output action to one that satisfies the

Table 7: This table provides an overview of the hyperparameters used in ARAM.

| Parameter | ARAM |
| --- | --- |
| Optimizer | Adam |
| Learning Rate | 0.0003 |
| Discount Factor | 0.99 |
| Replay Buffer Size | 1000000 |
| Number of Hidden Units per Layer | [256, 256] |
| Number of Samples per Minibatch | 256 |
| Nonlinearity | ReLU |
| Target Smoothing Coefficient | 0.005 |
| Target Update Interval | 1 |
| Gradient Steps | 1 |
| Sample Ratio for Augmented Replay Buffer ($\eta$) | 0.2 |
| Decay Interval for $\eta$ | 10,000 |
| Decay Factor for $\eta$ | 0.9 |

Table 8: This table provides an overview of the hyperparameters used in the baseline methods.

| Parameter | DPre+ | SPre+ | NFWPO | FlowPG |
| --- | --- | --- | --- | --- |
| Learning Rate | 0.001 | 0.001 | 0.001 | 0.001 |
| FW Learning Rate | - | - | 0.01 | - |
| Discount Factor | 0.99 | 0.99 | 0.99 | 0.99 |
| Replay Buffer Size | 1000000 | 1000000 | 1000000 | 1000000 |
| Number of Hidden Units | [256, 256] | [256, 256] | [256, 256] | [400, 300] |
| Number of Samples per Minibatch | 256 | 256 | 256 | 100 |
| Target Smoothing Coefficient | 0.005 | 0.005 | 0.005 | 0.005 |
| Target Update Interval | 1 | 1 | 1 | 1 |
| Gradient Steps | 1 | 1 | 1 | 1 |

constraints. An intuitive approach is through a QP operation to find the closest feasible action to the original one within the acceptable action space. We refer to this as the QP-Solver:

$$\text{QP-Solver}(s, a, \mathcal{C}(s)) = \arg\min_{a' \in \mathcal{C}(s)} ||a' - a||_2. \tag{12}$$

Kasaura et al. (2023) describe a family of algorithms based on the QP-Solver. However, as the complexity of the action space increases, the computation time for this type of approach can become lengthy. To ensure the feasibility of policy learning within a reasonable timeframe, it is essential to minimize reliance on the QP-Solver.

### E.2 FRANK-WOLFE POLICY OPTIMIZATION (FWPO)

Aside from the QP-Solver, Lin et al. (2021) employ the Frank-Wolfe (FW) method to update the policy and propose Frank-Wolfe Policy Optimization (FWPO). To describe the different policy parameters, FWPO uses $\phi_k$ to denote the policy parameters in the $k$-th iteration. To update the policy with the action constraints, FWPO adopts a generalized policy iteration framework (Sutton, 2018), which consists of two subroutines:

(i) Policy update via state-wise FW: For each state $s$ and the corresponding learning rate $\alpha_k(s)$, search for feasible action, then guide the current policy parameters update.

$$c_k(s) = \arg\max_{c \in \mathcal{C}(s)} \langle c, \nabla_a Q_\theta(s, a; \pi_{\phi_k})|_{a=\pi_{\phi_k}(s)} \rangle, \tag{13}$$

$$\pi_{\phi_{k+1}}(s) \leftarrow \pi_{\phi_k}(s) + \alpha_k(s)(c_k(s) - \pi_{\phi_k}(s)), \tag{14}$$

(ii) Evaluation of the current policy: Use any policy evaluation algorithm to obtain $Q_\theta(s, a; \pi_{\phi_{k+1}})$. Where $c_k(s) - \pi_{\phi_k}(s)$ is the update direction.

### E.3 FLOWPG

Brahmanage et al. (2023) adopt normalizing to create an invertible mapping between the support of a simple distribution and the space of valid actions. Specifically, the focus is on the conditional

RealNVP model, which is well-suited for the general ACRL setting where the set of valid actions depends on the state variable. The conditional RealNVP extends the original RealNVP by incorporating a conditioning variable in both the prior distribution and the transformation functions. These transformation functions, implemented as affine coupling layers, enable efficient forward and backward propagation during model learning and sample generation.

---

**Algorithm 2:** FlowPG Algorithm

**Input** : Initial parameter vectors $\theta, \phi, \psi$

1 Initialize replay buffer $\mathcal{B}$;
2 **for** *episode = 1, ..., M* **do**
3      Initialize the random noise generator $\mathcal{N}$ for action exploration;
4      **for** $t = 1, ..., T$ **do**
5          Select action $\tilde{a}_t = \pi_\phi(s_t) + \mathcal{N}_t$ based on current policy and exploration noise;
6          Apply flow and get the environment action $a_t = f_\psi(\tilde{a}_t, s_t)$;
7          **if** $a_t$ *is invalid* **then**
8             $a_t \leftarrow$ QP-Solver$(s_t, a_t, \mathcal{C}(s_t))$;
9          Update actor policy using the sampled policy gradient:

$$\nabla_\phi J(\pi_\phi) = \nabla_a Q_\theta(s_t, a; \pi) \nabla_{\tilde{a}} f_\psi(\tilde{a}, s_t) \nabla_\theta \pi_\phi(s_t)\big|_{\tilde{a}=\pi_\phi(s_t), a=f_\psi(\tilde{a}, s_t)}$$

---

**Learning the RL-Model.** Integrating DDPG with normalizing flows involves using the learned mapping directly in the original policy network, which improves training speed and stability. The architecture of the policy network consists of the original DDPG policy network and the learned mapping function $f_\psi$. The objective is to learn a deterministic policy $f_\psi(\pi_\phi(s), s)$ that gives the action $a$ given a state $s$, maximizing $J(\pi_\phi)$:

$$\max_\phi J(\pi_\phi) = \mathbb{E}_{s \sim \mathcal{B}}[Q_\theta(s, f_\psi(\pi_\phi(s), s); \pi_\phi)] \tag{15}$$

where $\mathcal{B}$ is the replay buffer and $\phi$ represents the Q-function parameters, treated as constants during policy update. The policy update involves gradient ascent with respect to the policy network parameters $\phi$:

$$\nabla_\phi J(\pi_\phi) = \mathbb{E}_{s \sim \mathcal{B}}[\nabla_a Q_\theta(s, a; \pi_\phi) \nabla_{\tilde{a}} f_\psi(\tilde{a}, s) \nabla_\phi \pi_\phi(s)\big|_{\tilde{a}=\pi_\phi(s), a=f_\psi(\tilde{a}, s)}] \tag{16}$$

The critic update follows the same rule as in DDPG, with actions stored in the replay buffer being either the flow model output or projected actions. The proposed method can be extended to other RL algorithms like SAC or PPO, as normalizing flows enable the computation of log probabilities of actions required during training. The pseudo code of FlowPG is provided in Algorithm 2.

