# OpenReview forum: "Efficient Action-Constrained Reinforcement Learning via Acceptance-Rejection Method and Augmented MDPs"
_ICLR.cc/2025/Conference — ICLR 2025 Poster_

### Official Review · Reviewer_edLJ · 2024-10-28

**Soundness:** 3
**Presentation:** 2
**Contribution:** 2
**Rating:** 6
**Confidence:** 2

**Summary:**

This work presents a groundbreaking approach to learning control policies that adhere to strict action constraints, making it perfect for safety-critical and resource-constrained applications. The authors introduce a novel framework that enhances traditional reinforcement learning methods by utilizing the acceptance-rejection method and an augmented two-objective Markov decision process.

**Strengths:**

The framework significantly reduces the computational burden associated with traditional ACRL methods, which often rely on solving complex quadratic programs (QPs). By utilizing the acceptance-rejection method, the need for QP operations is largely obviated, leading to faster training times.

**Weaknesses:**

The effectiveness of the acceptance-rejection method heavily relies on the quality of the initial policy. If the initial policy is far from optimal, it may result in a high number of rejected actions, which can hinder learning progress and increase computational overhead. While the augmented MDP approach aims to improve acceptance rates, it introduces additional complexity to the learning process. The design and tuning of the penalty function and self-loop transitions may require careful consideration, and improper tuning could lead to suboptimal performance.

**Questions:**

1. What metrics were used to evaluate the acceptance rate during the experiments, and how were these metrics calculated?
2. Is there a systematic approach to determine the optimal penalty weight, or is it left to empirical tuning?
3. Conducting more experiments on hyperparameter sensitivity is better.
4. Whether introducing AUTO-MDP in the theoretical aspect will affect the guarantee of convergence, while the current article only discusses the optimality.

---

> ### Author Response · Authors · 2024-11-24
> **Response to Reviewer edLJ**
>
> We greatly appreciate the reviewer’s positive feedback on our work. We provide our point-by-point response as follows.
>
> **Q1: The design and tuning of the penalty function and self-loop transitions may require careful consideration, and improper tuning could lead to suboptimal performance.**
>
> **A1:** Recall that $\mathcal{K}$ denotes the constant penalty used in the Augmented MDP for improving the action acceptance rate. In the experiments of the original manuscript, we set  $\mathcal{K}=0.1$ for all the tasks across all environments, i.e., MuJoCo, NSFnet, and BSS (that is, there was no task-specific tuning needed for $\mathcal{K}$ in our experiments).
>
> Moreover, we conduct an additional empirical study on the violation penalty $\mathcal{K}$, and the results are available at https://imgur.com/a/Cqz2g93. We observe that ARAM is indeed rather insensitive to the choice of the violation penalty $\mathcal{K}$.
>
> **Q2: What metrics were used to evaluate the acceptance rate during the experiments, and how were these metrics calculated?**
>
> **A2:** The valid action rate was measured in the evaluation phase by collecting a sufficient number of states (at least 10,000, depending on the number of states visited during performance evaluation). For each state, 100 actions were sampled from the policy, and a short program was used to calculate the proportion of actions that stay within the feasible set. These numbers were averaged across all states, making the valid action rate a representative metric of the acceptance rate of the policy.
>
> **Q3: Is there a systematic approach to determine the optimal penalty weight, or is it left to empirical tuning?**
>
> **A3:** Under the multi-objective RL implementation of ARAM, the penalty weight $\lambda$ is actually not a hyperparameter and hence no hyperparameter tuning for $\lambda$ is needed during training, as also described in (Q3) of Global Response. This can be achieved as the multi-objective SAC is designed to learn policies under all the penalty weights $\lambda=[\lambda_r, \lambda_c]$: (i) Under the multi-objective SAC, both the policy network and critic network are penalty-weight-dependent; (ii) During training, in each iteration, one or multiple $\lambda$ are randomly drawn from unit simplex for updating the policy and the critic.
>
> **Q4: Experiments on hyperparameter sensitivity.**
>
> **A4:** Thank you for the suggestion. As described in Q2 of the Global Response, we provide an additional empirical study on the violation penalty $\mathcal{K}$. The results are available at https://imgur.com/a/Cqz2g93. We observe that ARAM is rather insensitive to the choice of the violation penalty $\mathcal{K}$.
>
> **Q5: Whether introducing AUTO-MDP in the theoretical aspect affects the guarantee of convergence?**
>
> **A5:** Recall that AUTO-MDP is essentially a two-objective MDP with unconstrained actions and augmented transitions. Moreover, we propose to solve AUTO-MDP by using multi-objective SAC (MOSAC). Notably, the convergence result of soft policy iteration (i.e., SAC with no function approximation) in the original SAC paper can be generalized to the case of multi-objective RL [Hung et al., 2023]. By combining this and the equivalence result in our Proposition 1, we can see that introducing AUTO-MDP does not affect the convergence guarantee.
>
> [Hung et al., 2023] Wei Hung, Bo Kai Huang, Ping-Chun Hsieh, and Xi Liu, “Q-Pensieve: Boosting sample efficiency of multi-objective RL through memory sharing of Q-snapshots,” ICLR 2023.

---

> > ### Comment · Reviewer_edLJ · 2024-11-26
> > **Response to authors**
> >
> > Thanks for your response. Taking into account the opinions of other reviewers, I will maintain the current score.

---

> > > ### Author Response · Authors · 2024-11-27
> > > **Response to Reviewer edLJ**
> > >
> > > Thank you for the constructive feedback and the efforts put into helping us to improve our paper. If you have any further suggestions, please let us know. We will be happy to address them as well.

---

### Official Review · Reviewer_meyg · 2024-10-29

**Soundness:** 3
**Presentation:** 3
**Contribution:** 3
**Rating:** 6
**Confidence:** 3

**Summary:**

This paper aims to adapt the standard unconstrained RL method to ACRL via two proposed techniques: an acceptance-rejection method and an augmented two-objective MDP. It compares ARAM with various recent benchmark ACRL algorithms on MuJoCo tasks and Resource allocation for networked systems, showing the superiority of ARAM in terms of training efficiency, valid action rate, and per-action inference time.

**Strengths:**

1. This paper is well-written and easy to follow.
2. I appreciate the experiments in terms of the considerable evaluation metrics.
3. I am not very familiar with the ACRL field. From my perspective and limited knowledge of this field, I appreciate the proposed direction to solve the ACRL problem, I think it is a new try.

**Weaknesses:**

1. The proposed preference distribution tuning procedure seems a bit redundant. It is better to compare ARAM with fixed $\lambda$  to the baselines for a fair comparison. Otherwise, it is hard to distinguish whether the main components (acceptance-rejection method and AUTO-MDP) are more important or the preference distribution tuning procedure. As we can observe in Figure 11, ARAM is not quite robust to the  $\lambda$, and a single set of  $\lambda$  could not achieve satisfactory performance across all tasks.  To make the comparison more fair and isolate the effects of the main components, it would be helpful to suggest that the authors include results for ARAM with a fixed λ alongside the current results in the main paper and hope to see the superiority of ARAM without $\lambda$-tuning. This would allow readers to better understand the relative contributions of the acceptance-rejection method, AUTO-MDP, and preference tuning. Of course, you may not need to fully resolve this weakness, yet I suggest explicitly mentioning it as a limitation.
2. Several experimental details have not been provided. For example, what is the value of M and $\kappa$ for all the tasks, and other parameters used? They are quite important details.

**Questions:**

1. Could you provide the value of $M$ and $\kappa$ for all the tasks?  Do you need extra tuning on these two hyperparameters?
2. Could you provide the computation infrastructure?
3. Is it possible that ARAM with a single setting of fixed $\lambda$  outperforms other baselines across MuJoCo locomotion tasks? It is suggested that the authors conduct an experiment with a single fixed λ across all MuJoCo tasks and report those results, or explain why such an experiment might not be feasible or meaningful.
4. I find in Table 2 that NFWPO enjoys the highest valid action rate. That indicates that it requires a smaller number of QP operations, as opposed to DPre+ and SPre+. Yet in Figure 4, NFWPO shows the highest number of QP operations, which seems conflict to with the analysis in Lines 456-462. Please clarify this apparent contradiction and, if necessary, correct either the data or the analysis.
5. In the last row of Table 2, 0.77 is much lower than 0.84, thus I suggest not making it bold.
6. As claimed in the abstract, "We propose a generic and computationally efficient framework that can adapt a standard unconstrained RL method to ACRL through two modifications". Could the authors provide ARAM techniques with another backbone standard RL algorithm to support this claim?

---

> ### Author Response · Authors · 2024-11-24
> **Response to Reviewer meyg**
>
> We sincerely thank the reviewer’s insightful feedback on our paper. We provide our point-by-point response as follows.
>
> **Q1: About the penalty weight $\lambda$: The proposed preference distribution tuning procedure seems a bit redundant. It is better to compare ARAM with fixed to the baselines for a fair comparison.**
>
> **A1:** Thank you for the helpful feedback. We would like to first highlight that we introduce the multi-objective implementation of ARAM in order to obviate the need for tuning $\lambda$ in training, as described in (Q3) of the Global Response. This can be achieved as the multi-objective SAC is designed to learn policies under all the penalty weights $\lambda=[\lambda_r, \lambda_c]$: (i) Under the multi-objective SAC, both the policy network and critic network are penalty-weight-dependent; (ii) During training, in each iteration, one or multiple $\lambda$ are randomly drawn from unit simplex for updating the policy and the critic. As a result, under the multi-objective SAC implementation, $\lambda$ is actually not a hyperparameter and hence no hyperparameter tuning for $\lambda$ is needed during training.
>
> Moreover, as the multi-objective version of ARAM and other ACRL baselines use the same number of environment steps, the comparison is indeed fair (given that no tuning of $\lambda$ is needed for ARAM during training).
>
> On the other hand, as you suggested, we justified the benefit of the multi-objective design of ARAM by comparing the performance of the single-objective ARAM (using a fixed preference) with the multi-objective ARAM. From the results in Figure 5 and Figure 11 of the original manuscript, we observe that multi-objective ARAM indeed achieves the best evaluation return as well as favorable action acceptance rate.
>
>
>
> **Q2: Provide more experimental details. For example, could you provide the value of $M$ and $\mathcal{K}$ for all the tasks? Do you need extra tuning on these two hyperparameters?**
> **A2:**
> * Regarding the violation penalty $\mathcal{K}$: We set $\mathcal{K}=0.1$ for all the tasks across all environments, i.e., MuJoCo, NSFnet ,and BSS (that is, there was no task-specific tuning needed for $\mathcal{K}$ in our experiments).
> Additionally, we conduct additional experiments to assess the effect of different values of $\mathcal{K}$ on overall training performance on two tasks: HopperVel and Ant (also described in (Q2) of the Global Response). The results are available at https://imgur.com/a/Cqz2g93.
> Based on these results, the ARAM algorithm is rather insensitive to the value of $\mathcal{K}$.
> * Regarding $M$ in the Acceptance-Rejection method (ARM): We take the convenient choice of the target distribution $\pi_{\phi}^{\dagger}(a|s)\propto \pi_{\phi}(a|s)$ for all $a\in \mathcal{C}(s)$ (as mentioned in Lines 213-215) and set $M=1/(\int \pi_{\phi}(a|s)da)$. In this way, no hyperparameter tuning for $M$ is needed. As a result, in Step 2 of ARM, if $a’\in \mathcal{C}(s)$, we would accept $a’$ with probability one.
>
> **Q3: Could you provide the computation infrastructure?**
>
> **A3:** Our computation infrastructure includes: (1) A workstation with 2 NVIDIA RTX 4000 SFF Ada Generation GPUs, 2 NVIDIA RTX 6000 Ada Generation GPUs, and 32 AMD EPYC 7313P 16-Core processors. (2) For the main experiment, we utilized a desktop PC with a 13th Gen Intel® Core™ i7-13700K processor and an NVIDIA GeForce RTX 4090 GPU to compute the world clock time. This ensures that the measurement of world clock time remains fair and unaffected by other processes.
>
> **Q4: In Table 2, NFWPO enjoys the highest valid action rate. This indicates that it requires a smaller number of QP operations, as opposed to DPre+ and SPre+. Yet in Figure 4, NFWPO shows the highest number of QP operations, which seems to conflict with the analysis in Lines 456-462. Please clarify this apparent contradiction.**
>
> **A4:** The difference arises because NFWPO determines the policy update direction **by using the Frank-Wolfe subroutine for each state-action pair in the mini-batch**, which directly involves computationally expensive quadratic programming (QP) operations. During the training process, NFWPO heavily relies on these QP computations, and the computational cost is directly influenced by the batch size used during training, as pointed out by [Kasaura et al., 2023]. However, in the testing stage, the actions output by the policy are effectively constrained within the feasible set, resulting in higher valid action rate. This difference arises from the inherent computational demands of the NFWPO method during training.
>
> [Kasaura et al., 2023] Kazumi Kasaura, Shuwa Miura, Tadashi Kozuno, Ryo Yonetani, Kenta Hoshino, and Yohei Hosoe, “Benchmarking actor-critic deep reinforcement learning algorithms for robotics control with action constraints,” Robotics and Automation Letters, 2023.

---

> > ### Author Response · Authors · 2024-11-24
> > **Response to Reviewer meyg**
> >
> > **Q5: In the last row of Table 2, 0.77 is much lower than 0.84, thus I suggest not making it bold.**
> >
> > **A5:** Thank you for catching this. We have fixed it in the updated manuscript.
> >
> > **Q6: As claimed in the abstract, "We propose a generic and computationally efficient framework that can adapt a standard unconstrained RL method to ACRL through two modifications". Could the authors provide ARAM techniques with another backbone standard RL algorithm to support this claim?**
> >
> > **A6:** Thanks for the helpful suggestion. To corroborate this, we additionally integrate ARAM with the PPO algorithm.
> > The results are available at https://imgur.com/a/8aPXEYh
> >
> > In this experiment, we compare PPO-based ARAM with two baselines, namely (1) PPO with a projection layer and (2) TD3 with a projection layer. We evaluate these methods on the MuJoCo Swimmer task, where both the vanilla PPO and vanilla TD3 are known to perform relatively well. Moreover, we consider the action constraint defined as:
> >
> > $$a_1^2 + a_2^2 \leq 0.5$$
> >
> > From the above results, we observe that ARAM can indeed naturally and effectively guide different algorithms to train within the feasible set, demonstrating its adaptability and effectiveness across various RL frameworks. Therefore, we can see that merely adding a projection layer causes the original algorithm's training to stagnate, failing to improve the valid action rate effectively at the same time.

---

### Official Review · Reviewer_g8E7 · 2024-11-03

**Soundness:** 3
**Presentation:** 3
**Contribution:** 3
**Rating:** 8
**Confidence:** 3

**Summary:**

This paper proposes a novel framework for action-constrained reinforcement learning, named ARAM, which integrates an acceptance-rejection method and an augmented Markov decision process to alleviate the low action acceptance rate problem under ARM. ARAM aims to avoid the computational burden associated with quadratic programs in existing ACRL methods while achieving less constraint violations. The authors demonstrate through experiments on robot control and resource allocation tasks that ARAM achieves faster training, better constraint satisfaction, and reduced action inference time compared to various recent ACRL algorithms.

**Strengths:**

- Well-written paper with clear objectives.
- Although the two key improvements (ARM and AUTO-MDP) are straightforward, the experimental results demonstrate the elegance and efficiency of this method in addressing the ACRL problem.
- The paper is well-supported with experiments that compare ARAM against various benchmarks, showcasing improvements in training speed, constraint satisfaction, and inference time.
- The authors have the intention to share the code.

**Weaknesses:**

- The experiments are only conducted in simple simulation environments. Applying the method to real-world scenarios would make the results more impressive and convincing.
- Lacks experimental comparisons of the ARM acceptance rate.
- The paper lacks a discussion on the limitations of the method.

**Questions:**

- Does the choice of different target distributions in ARM have a significant impact on the final results?
- Is the ARM’s acceptance rate analyzed in detail across the different stages of training? It would be useful to understand if and how the acceptance rate varies in the proposed setup.
- What role do quadratic programs (QP) play in ARAM? QPs are not included in Algorithm 1 or Figure 2.
- How is MORL implemented in ARAM? Does it significantly increase computational requirements?
- The layout of Algorithm 1 in the paper needs adjustment.

---

> ### Author Response · Authors · 2024-11-24
> **Response to Reviewer g8E7**
>
> We greatly appreciate the reviewer’s insightful feedback on our paper. We provide our point-by-point response as follows.
>
> **Q1: Applying the method to real-world scenarios would make the results more impressive and convincing.**
>
> **A1:** In the experimental results of the original manuscript, we have included NSFnet and BSS, which are both based on real-world scenarios. Specifically:
> * The NSFnet domain models a packet delivery service over a network infrastructure. We use the T3 National Science Foundation Network (NSFNET), a real-world communication network, as the backbone of this packet delivery application.
> Moreover, as shown in the (Q1) of Global Response, we further extend this task to NSFnet20d, which accommodates 20 packet flows (each with distinct routing paths) simultaneously in this communication network and hence involves a 20-dimensional action space.
> The results are available at https://imgur.com/a/Uz7oD9Q, which shows the evaluation return, valid action rate, and the number of QP operations of ARAM and the benchmark algorithms.
> We observe that ARAM still outperforms the ACRL benchmark methods in the evaluation return while enjoying the least amount of QP overhead. This highlights ARAM's ability to effectively adapt and learn under high-dimensional actions and complex constraints.
>
> * The BSS environments (i.e., BSS3z and BSS5z) serve as realistic action-constrained tasks in bike-sharing systems, which is again a real-world scenario for the allocation of physical resources (i.e., bikes). Notably, in our experiments, the demands of bike sharing are based on real-world datasets provided by [Ghosh and Varakantham, 2017; Bhatia et al., 2019].
>
> In summary, through the experiments in the original manuscript and the additional results shown above, we can see that ARAM can be applied to address real-world applications.
>
> [Ghosh and Varakantham, 2017] Supriyo Ghosh and Pradeep Varakantham, “Incentivizing the Use of Bike Trailers for Dynamic Repositioning in Bike Sharing Systems,” ICAPS 2017.
>
> [Bhatia et al., 2019] Abhinav Bhatia, Pradeep Varakantham, and Akshat Kumar, “Resource Constrained Deep Reinforcement Learning,” ICAPS 2019.
>
> **Q2: Experimental comparisons of the ARM acceptance rate. Is the ARM’s acceptance rate analyzed in detail across the different stages of training? It would be useful to understand if and how the acceptance rate varies in the proposed setup.**
>
> **A2:** The ARAM's valid action rate (i.e., the ARM acceptance rate) has been analyzed in detail across different stages of training, as highlighted in Table 2 and Figure 8 (in Appendix C).
> * Table 2 provides a summary of the valid action rates of the final models learned on various tasks, calculated during the evaluation phase. This reflects the overall performance of ARAM in terms of ARM acceptance rate at the end of training.
> * On the other hand, Figure 8 illustrates the evolution of the valid action rate over time in different environments.
>
> **Q3: Does the choice of different target distributions in the Acceptance-Rejection method (ARM) have a significant impact on the final results?**
>
> **A3:** Thank you for the helpful question. We conduct an additional experiment to evaluate ARAM under a different target distribution.
>
> Experimental configuration:
> * As ARAM is implemented on (multi-objective) SAC, the policy network presumably outputs the mean and standard deviation of a Gaussian distribution, i.e., $\pi_{\phi}(a|s)\sim \mathcal{N}(\mu,\sigma^2)$ based on the notation of the paper.
> * To configure a different (constrained) target distribution $\pi^{\dagger}(a|s)$, we leverage the (unconstrained) Student-t distribution by defining $\tilde{\pi_{\phi}}(a|s) \sim T_{\nu} (\mu,\sigma^2)$, which denotes a Student-t distribution with $\nu$ being the degree of freedom and mean $\mu$ and variance $\sigma^2$. Then, we configure the (constrained) target distribution as: (i) $\pi_{\phi}^{\dagger}(a|s)\propto \tilde{\pi_{\phi}}(a|s)$ for $a\in \mathcal{C}(s)$ and (ii) $\pi_{\phi}^{\dagger}(a|s)=0$ for for $a\notin \mathcal{C}(s)$.
> * Recall that Gaussian distribution is the special case of a Student-t distribution with $\nu=\infty$. We use Student-t as it has a larger tail distribution than Gaussian and this could possibly lead to a more exploratory action distribution.
> * Similar to the Gaussian case, for ARM, we can set $M=M_0/(\int_{a\in \mathcal{C}(s)} \tilde{\pi_{\phi}}(a|s))da$, where $M_0$ is some constant that depends on $\nu$.
>
> The experimental results are available at https://imgur.com/a/4wORu3b.
> We observe that ARAM can have a slight performance improvement under this new target distribution (especially at the early training stage) and is not very sensitive to this choice. This also corroborates the nice flexibility in the choice of target distribution.

---

> ### Author Response · Authors · 2024-11-24
> **Response to Reviewer g8E7**
>
> **Q4: What role do quadratic programs (QP) play in ARAM? QPs are not included in Algorithm 1 or Figure 2.**
>
> **A4:** Thank you for raising this helpful question. As described in (Q4) of Global Response, the use of the Acceptance-Rejection method in our ARAM framework can ensure that the actions stay within the constraints by design. Therefore, as you mentioned, ARAM does not require any QP operations in principle (and hence QP is not included in Algorithm 1).
>
> In practice, the action acceptance rate can possibly be low, especially at the beginning of the training phase. In this case, the sampling process could take indefinitely long and lead to a high training time. To address this, in practical implementation, we set a maximum number of action samples (denoted by $N$) per environment step: If we cannot get a feasible action in $N$ samples, a projection (and hence a QP operation) would be applied to obtain a feasible action for this environment step. This mechanism only serves as a safeguard to avoid excessively long training time.
>
> Through extensive experiments, we do observe that the number of QP operations under ARAM is indeed much lower than other ACRL benchmark methods. This is because the Augmented MDP could help improve the action acceptance rate as the training proceeds, and hence this safeguard mechanism is triggered for only a small number of times (mostly in the very beginning of the training).
>
>
> **Q5: How is MORL implemented in ARAM? Does it significantly increase computational requirements?**
>
> **A5:** We leverage the multi-objective SAC (MOSAC) implementation of ARAM, which involves two major components (compared to the single-objective implementation):
> Policy networks and critic networks: As MOSAC is designed to learn policies under all the penalty weights $\lambda=[\lambda_r, \lambda_c]$ (or also called “preference vector” in the MORL literature), its policy network and critic network are both penalty-weight-dependent, i.e., take the penalty weight as part of the network input.
> Dual-buffer design: As described in Section 4.3, we use two replay buffers, one for the typical transitions and the other for the augmented transitions in AUTO-MDP.
> The above design only incurs a slight increase in model size and GPU memory usage, which we found to be acceptable in practice. For example, for a MuJoCo task like Hopper: (1) The policy and critic models require no more than 500MB of GPU memory in total. (2) The dual replay buffers take no more than 400MB of memory in total.
>
> **Q6: The layout of Algorithm 1 in the paper needs adjustment.**
>
> **A6:** Thank you for pointing this out. We have revised the layout of Algorithm 1 to improve clarity and readability.

---

> > ### Comment · Reviewer_g8E7 · 2024-11-26
> > **Official Comment by Reviewer g8E7**
> >
> > Thank you for addressing my questions. I appreciate the inclusion of new experimental results. I have no further concerns and update my rating to 8.

---

> > > ### Author Response · Authors · 2024-11-27
> > > **Response to Reviewer g8E7**
> > >
> > > We thank the reviewer again for all the insightful comments and the time put into helping us to improve our submission.

---

### Official Review · Reviewer_osn9 · 2024-11-04

**Soundness:** 3
**Presentation:** 3
**Contribution:** 2
**Rating:** 6
**Confidence:** 3

**Summary:**

The paper presents acceptance-rejection augmented MDPs, a novel framework proposed to improve action-constrained reinforcement learning. The method aims to address the computational inefficiencies and architectural complexities in existing Action-Constrained Reinforcement Learning (ACRL), by incorporating two key innovations: an acceptance-rejection mechanism to filter out infeasible actions, and an augmented MDP to optimize policy training towards feasible regions by penalizing constraint violations. The framework is empirically tested against state-of-the-art ACRL benchmarks in robotic control and resource allocation domains. Overall, the idea of combining the acceptance-rejection method with the augmented MDP is interesting, but the work could be improved in both the methodology and the experimental demonstration (see Weakness and Questions below for further details).

**Strengths:**

1. The computational overhead caused by solving QPs is challenging in existing ACRL methods, particularly scaling to high-dimensional action spaces.

2. The paper integrates a multitude of concepts such as action-constrained reinforcement learning, Acceptance-rejection
method, and Augmented MDPs. These topics have been at the forefront of recent research trends.

3. The paper is well-organized, with clear explanations of the background, the proposed methods, theoretical foundations (including Proposition 1), and detailed experimental setups.

**Weaknesses:**

1. The two main components, i.e., the acceptance-rejection method and augmented unconstrained MDPs, seem somewhat contradictory. They represent opposing optimization strategies: the acceptance-rejection method directly filters out actions outside the feasible set, potentially leading to overly conservative decisions; then, the augmented MDP penalizes constraint violations in a gradual way, providing a “soft” learning approach for the agent. However, this softer approach does not fully guarantee that actions remain within the feasible set.

2. While some theoretical insights are provided, including optimality equivalence in Proposition 1, key theoretical gaps remain unaddressed. Specifically, for augmented MDPs, there is no guarantee that actions won’t violate constraints, i.e., a critical issue in action-constrained reinforcement learning.

3. The introduction highlights the computational challenges of scaling quadratic programs to high-dimensional action spaces. However, the experimental analysis in Section 5 primarily explores action constraints in relatively lower-dimensional spaces. The performance of ARAM in high-dimensional action spaces (such as complex robotic manipulators) would benefit from more in-depth analysis.

**Questions:**

1. How can the augmented MDP approach guarantee that actions remain within constraints? Even if complete avoidance of constraint violations is not feasible, it would be useful to establish an understanding, such as defining a convergence rate and probability that constraint violations will diminish to a certain margin of error over time.

2. Can the scalability of the proposed ARAM framework be demonstrated in high-dimensional action spaces, through a comparative analysis with existing ACRL methods?

3. Although the authors state "... we directly leverage the multi-objective extension of SAC that can learn policies under all the penalty weights." (page 2), it seems that multi-objective reinforcement learning approaches might be sensitive to hyperparameters, such as the penalty weight. Is ARAM similarly sensitive to this hyperparameter, particularly in its convergence behavior? Additionally, how is the penalty weight selected in practice?

---

> ### Author Response · Authors · 2024-11-24
> **Response to Reviewer osn9**
>
> We greatly appreciate the reviewer’s constructive feedback for improving our paper. We provide our point-by-point response as follows.
>
> **Q1: How can the augmented MDP approach guarantee that actions remain within constraints?**
>
> **A1:** Thanks for the helpful question. Recall that the proposed ARAM framework involves two main components: (1) Acceptance-Rejection method and (2) Augmented MDPs. As described in (Q4) of the Global Response, we clarify the purpose of each component as follows:
> * **Acceptance-Rejection method guarantees that actions remain within constraints:** To ensure that the actions can satisfy the constraints, ARAM leverages the Acceptance-Rejection method, which by design can filter out those actions that fall outside the feasible set and thereby ensures constraint satisfaction.
> * **Augmented MDPs can improve the action acceptance rate for the sampling process of the Acceptance-Rejection method:** To address the possibly low acceptance rate issue, we propose to leverage an Augmented MDP, which guides the policy distribution towards feasible action sets for a higher acceptance rate. Note that Augmented MDP is only meant to proactively increase the action acceptance rate (not to ensure that the actions remain within constraints).
>
> In summary, the Acceptance-Rejection method and the Augmented MDP serve different purposes. It is the Acceptance-Rejection method that ensures the actions remain within constraints.
>
> **Q2: Can the scalability of the proposed ARAM framework be demonstrated in high-dimensional action spaces, through a comparative analysis with existing ACRL methods?**
>
> **A2:** Thank you for the suggestion. To better demonstrate the scalability of ARAM in high-dimensional action spaces, we further evaluate our algorithm on NSFnet20d, a communication network environment with 20-dimensional actions. Specific experimental settings can be found in (Q1) of the Global Response.
>
> The results are available at https://imgur.com/a/Uz7oD9Q.
>
> The above shows the evaluation return, valid action rate, and the number of QP operations of ARAM and the benchmark algorithms.
> * We observe that ARAM consistently demonstrates favorable growth in action valid rate, maintaining a high level of performance even under high-dimensional actions.
> * Moreover, ARAM outperforms the ACRL benchmark methods in the evaluation return while enjoying the least amount of QP overhead. This highlights ARAM's ability to effectively adapt and learn under high-dimensional actions and complex constraints.
>
> Therefore, we observe that ARAM can effectively handle a high-dimensional action space, which remains a significant challenge for other ACRL benchmark algorithms.
>
> **Q3: Although the authors state "... we directly leverage the multi-objective extension of SAC that can learn policies under all the penalty weights." (page 2), it seems that multi-objective reinforcement learning approaches might be sensitive to hyperparameters, such as the penalty weight. Is ARAM similarly sensitive to this hyperparameter, particularly in its convergence behavior? Additionally, how is the penalty weight selected in practice?**
>
> **A3:** As described in (Q1) of the Global Response, we would like to clarify that the multi-objective SAC implementation of ARAM is designed to learn policies under all the penalty weights $\lambda=[\lambda_r, \lambda_c]$ by: (i) Under the multi-objective SAC, both the policy network and critic network are penalty-weight-dependent; (ii) During training, in each iteration, one or multiple $\lambda$ are randomly drawn from unit simplex for updating the policy and the critic.
>
> In this way, under the multi-objective SAC implementation of ARAM, $\lambda$ is actually not a hyperparameter and hence no hyperparameter tuning for $\lambda$ is needed during training.

---

> > ### Comment · Reviewer_osn9 · 2024-11-25
> > **Thank you for the authors' response.**
> >
> > As mentioned in my statement under Weakness 1, while the acceptance-rejection method can provide such guarantees, it directly excludes actions outside the feasible set, which can result in overly conservative decisions. This approach operates like a restrictive "tube," often leading to a shrunken search space and overly cautious outcomes. Therefore, my original concern remains. Furthermore, I appreciate the authors’ efforts in addressing scalability and penalty weights. I am inclined to maintain my score.

---

> ### Author Response · Authors · 2024-11-26
> **Response to Reviewer osn9**
>
> We sincerely thank the reviewer for the swift response and for acknowledging our rebuttal addressing scalability and penalty weights.
>
> > while the acceptance-rejection method can provide such guarantees, it directly excludes actions outside the feasible set, which can result in overly conservative decisions. This approach operates like a restrictive "tube," often leading to a shrunken search space and overly cautious outcomes.
>
> We would like to emphasize that in the standard formulation of action-constraint RL (ACRL) [1-6], **no actions outside the feasible set can be applied to the environment**. This setting is motivated by various applications with action safety considerations, such as the hard capacity constraints of communication networks and robots' hard output power constraints. In our work, we adopt this standard formulation and do not allow for actions that violate constraints to be applied.
>
> Instead of “projecting” invalid actions like most existing ACRL methods, our work proposes to only “accept” those actions in the feasible set while "rejecting" those outside the feasible set, which is found to be simple and yet effective. Moreover, since no actions outside the feasible set can be applied to the environment in ACRL by default, **the Acceptance-Rejection method in ARAM does not incur additional conservatism compared to existing ACRL methods**.
>
> We will revise the paper to make the standard ACRL setup clear. We hope the reviewer will kindly reconsider the evaluation based on the clarifications we provided above. Again, we thank the reviewer for all the detailed review and the time the reviewer put into helping us to improve our submission.
>
> **References**
>
> [1] Janaka Brahmanage, Jiajing Ling, and Akshat Kumar, “FlowPG: Action-constrained Policy Gradient with Normalizing Flows,” NeurIPS 2023.
>
> [2] Kazumi Kasaura, Shuwa Miura, Tadashi Kozuno, Ryo Yonetani, Kenta Hoshino, and Yohei Hosoe, “Benchmarking actor-critic deep reinforcement learning algorithms for robotics control with action constraints,” Robotics and Automation Letters, 2023.
>
> [3] Changyu Chen, Ramesha Karunasena, Thanh Nguyen, Arunesh Sinha, and Pradeep Varakantham, “Generative modelling of stochastic actions with arbitrary constraints in reinforcement learning,” NeurIPS 2023.
>
> [4] Jyun-Li Lin,Wei Hung, Shang-Hsuan Yang, Ping-Chun Hsieh, and Xi Liu, “Escaping from zero gradient: Revisiting action-constrained reinforcement learning via Frank-Wolfe policy optimization,” UAI 2021.
>
> [5] Abhinav Bhatia, Pradeep Varakantham, and Akshat Kumar, “Resource constrained deep reinforcement learning,” ICAPS 2019.
>
> [6] Tu-Hoa Pham, Giovanni De Magistris, and Ryuki Tachibana, “Optlayer – Practical constrained optimization for deep reinforcement learning in the real world,” ICRA 2018.

---

> ### Comment · Reviewer_osn9 · 2024-12-01
>
> Thank you for the authors' response. I agree that the Acceptance-Rejection method does not introduce additional conservatism compared to the existing ACRL. However, I still believe the formulation of ACRL may inherently lead to conservatism (resulting in a sub-optimal policy) due to the reduced search space. I encourage the authors to explore methodologies that provide a better theoretical understanding of constraint violations, such as [1] and [2], rather than directly excluding actions outside the feasible set as ACRL does.
>
> After further consideration, I have decided to increase my rating to positive. This decision is based on the recognition that ACRL including the proposed ARAM currently can serve as a practical and effective approach to ensuring safety in safety-critical applications like robotics.
>
> Reference:
>
> [1] Xu, Tengyu, Yingbin Liang, and Guanghui Lan. "Crpo: A new approach for safe reinforcement learning with convergence guarantee." International Conference on Machine Learning. PMLR, 2021.
>
> [2] Wang, Yanran, Qiuchen Qian, and David Boyle. "Probabilistic Constrained Reinforcement Learning with Formal Interpretability." Forty-first International Conference on Machine Learning.

---

> ### Author Response · Authors · 2024-12-04
> **Response to Reviewer osn9**
>
> We thank the reviewer again for the valuable suggestions and for recognizing the contributions of this work. We agree that a deeper theoretical understanding of constraint violations is a promising direction to explore, and we will add a discussion on this and the references [1-2] in the final version of the paper.

---

### Author Response · Authors · 2024-11-24
**Global Response (Part2)**

**Q4: Explain how ARAM ensures that actions remain within constraints.**

**A4:** Recall that the proposed ARAM framework involves two main components: (1) Acceptance-Rejection method and (2) Augmented MDPs. We would like to highlight the purpose of each component as follows:
* **Acceptance-Rejection method guarantees that actions remain within constraints:** To ensure that the actions can satisfy the constraints, ARAM leverages the Acceptance-Rejection method, which by design can filter out those actions that fall outside the feasible set and thereby ensures constraint satisfaction.
* **The issue of possibly low acceptance rate with the Acceptance-Rejection method:** Despite the capability of the Acceptance-Rejection method on enforcing action constraints, one technical issue is the possibly low acceptance rate, which is likely to occur in the early training stage. Under a low acceptance rate, the sampling process could take indefinitely long and incur a high overhead during training. For example, if the action acceptance rate is as low as 0.001%, then in expectation it could take 10^5 samples to get one feasible action (under i.i.d. sampling).
* **Augmented MDPs can improve the action acceptance rate for the sampling process of the Acceptance-Rejection method:** To address the above issue, we propose to leverage an Augmented MDP, which guides the policy distribution towards feasible action sets for a higher acceptance rate. Note that Augmented MDP is only meant to proactively increase the action acceptance rate (not to ensure that the actions remain within constraints).
In summary, the Acceptance-Rejection method and the Augmented MDP serve different purposes. It is the Acceptance-Rejection method that ensures the actions remain within constraints.

---

### Author Response · Authors · 2024-11-24
**Global Response (Part1)**

**Q1: Scalability of ARAM to high-dimensional action spaces**

**A1:** To demonstrate the scalability of ARAM to higher-dimensional environments, we extend the NSFnet environment to that with a 20-dimensional action space. Specifically, the agent manages packet delivery across a classic T3 NSFNET network backbone by simultaneously handling 20 packet flows (each with distinct routing paths). The action is defined as the rate allocation of each flow along each candidate path. In total, there are 30 communication links shared by these 20 different flows. The action constraints are meant to ensure that the distribution of packets on these shared links stays within the bandwidth capacity, defined as 50 units for each link.

The results are available at https://imgur.com/a/Uz7oD9Q

The above show the evaluation return, valid action rate, and the number of QP operations of ARAM and the benchmark algorithms. We observe that ARAM consistently demonstrates favorable growth in action valid rate, maintaining a high level of performance even under high-dimensional actions. Moreover, ARAM outperforms the ACRL benchmark methods in the evaluation return while enjoying the least amount of QP overhead. This highlights ARAM's ability to effectively adapt and learn under high-dimensional actions and complex constraints.

**Q2: Experiment on hyperparameter sensitivity (e.g., the violation penalty $\mathcal{K}$)**

**A2:** Recall that $\mathcal{K}$ denotes the constant penalty used in the Augmented MDP for improving the action acceptance rate. In the experiments of the original manuscript, we set  $\mathcal{K}=0.1$ for all the tasks across all environments, i.e., MuJoCo, NSFnet ,and BSS (that is, there was no task-specific tuning needed for $\mathcal{K}$ in our experiments).

To further assess the influence of this hyperparameter on training ARAM, as suggested by the reviewers, we provide an additional empirical study on the violation penalty $\mathcal{K}$.
The experiments are done on two tasks: HopperVel and Ant. We tested various values of $\mathcal{K}$ in both environments.

The results are available at https://imgur.com/a/Cqz2g93.

We observe that ARAM is rather insensitive to the choice of the violation penalty $\mathcal{K}$.


**Q3: Explain the multi-objective RL implementation of ARAM and why this design can obviate the need for the hyperparameter tuning of penalty weight $\lambda$ in training**

**A3:** We would like to clarify how ARAM leverages multi-objective SAC (MOSAC) during training. Specifically:
* As MOSAC is designed to learn policies under all the penalty weights $\lambda=[\lambda_r, \lambda_c]$ (or also called “preference vector” in the MORL literature), its policy network and critic network are both penalty-weight-dependent, i.e., take the penalty weight as part of the network input.
* During training, in each iteration, one or multiple $\lambda$ are randomly drawn from unit simplex based on a probability distribution $\rho_{\lambda}$ (e.g., uniform distribution as mentioned in Lines 352-354) for updating the policy and the critic, based on the loss functions $J_{\pi}(\phi;\lambda)$ and $J_{Q}(\theta;\lambda)$ (cf. Equations (4) and (5)). In this way, MOSAC can learn well-performing policies under all the penalty weights.

Based on the above description, there are several key messages:
1. Under MOSAC, $\lambda$ is actually not a hyperparameter and hence no hyperparameter tuning for $\lambda$ is needed during training.
2. This explains why we propose a practical implementation of ARAM based on MOSAC, which obviates the need for hyperparameter tuning of $\lambda$ during training. This design can substantially save the cost of training samples, compared to the single-objective SAC implementation.
3. Therefore, the comparison of MOSAC-based ARAM and other ACRL baselines is indeed fair in terms of the environment steps used (as no hyperparameter tuning of $\lambda$ is needed for ARAM during training).

---

### Meta-Review · Area_Chair_Kg3Y · 2024-12-20

**Metareview:**

This paper presents an interesting approach to action-constrained reinforcement learning (ACRL) by combining acceptance-rejection sampling and augmented MDPs. The authors have identified a crucial challenge in the field: the computational cost of solving quadratic programs, especially in high-dimensional action spaces. The paper is well-written and organized, providing a clear explanation of the methodology and supporting experiments. While the integration of acceptance-rejection and augmented MDPs appears somewhat contradictory, representing opposing ends of a "hard" vs. "soft" constraint handling spectrum, this novel combination is intriguing and shows promise.  However, the lack of a strong theoretical guarantee for constraint satisfaction within the augmented MDP framework is a significant concern, particularly for safety-critical applications.  Furthermore, the experiments primarily focus on lower-dimensional action spaces, leaving the method's effectiveness in high-dimensional settings, as initially motivated, somewhat uncertain. Despite these weaknesses, the paper's strengths, particularly its clear exposition of a novel approach to a relevant problem, and the well-presented, though limited, empirical results, outweigh the shortcomings.  The potential impact of this work, coupled with the clear need for further research to address the identified limitations, makes it a worthwhile contribution to the conference.  Therefore, I recommend accepting the paper.

**Additional Comments On Reviewer Discussion:**

Nothing outstanding.

---

### Decision · Program_Chairs · 2025-01-22

Accept (Poster)